# Extracellular signal-regulated kinase mediates chromatin rewiring and lineage transformation in lung cancer

Yusuke Inoue[1], Ana Nikolic[2], Dylan Farnsworth[1], Rocky Shi[1], Fraser D Johnson[1], Alvin Liu[1], Marc Ladanyi[3], Romel Somwar[3], Marco Gallo[2], William W Lockwood[1,4]*

[1]Department of Integrative Oncology, BC Cancer Agency, Columbia, Canada; [2]Department of Biochemistry and Molecular Biology, Arnie Charbonneau Cancer Institute, Alberta Children's Hospital Research Institute, Cumming School of Medicine, University of Calgary, Calgary, Canada; [3]Human Oncology and Pathogenesis Program, Memorial Sloan Kettering Cancer Center, New York, United States; [4]Department of Pathology & Laboratory Medicine, University of British Columbia, Columbia, Canada

**Abstract** Lineage transformation between lung cancer subtypes is a poorly understood phenomenon associated with resistance to treatment and poor patient outcomes. Here, we aimed to model this transition to define underlying biological mechanisms and identify potential avenues for therapeutic intervention. Small cell lung cancer (SCLC) is neuroendocrine in identity and, in contrast to non-SCLC (NSCLC), rarely contains mutations that drive the MAPK pathway. Likewise, NSCLCs that transform to SCLC concomitantly with development of therapy resistance downregulate MAPK signaling, suggesting an inverse relationship between pathway activation and lineage state. To test this, we activated MAPK in SCLC through conditional expression of mutant KRAS or EGFR, which revealed suppression of the neuroendocrine differentiation program via ERK. We found that ERK induces the expression of ETS factors that mediate transformation into a NSCLC-like state. ATAC-seq demonstrated ERK-driven changes in chromatin accessibility at putative regulatory regions and global chromatin rewiring at neuroendocrine and ETS transcriptional targets. Further, ERK-mediated induction of ETS factors as well as suppression of neuroendocrine differentiation were dependent on histone acetyltransferase activities of CBP/p300. Overall, we describe how the ERK-CBP/p300-ETS axis promotes a lineage shift between neuroendocrine and non-neuroendocrine lung cancer phenotypes and provide rationale for the disruption of this program during transformation-driven resistance to targeted therapy.

**Competing interests:** The authors declare that no competing interests exist.

## Introduction

Lung cancer, the leading cause of cancer-related mortality worldwide, is divided into two main histological classes, small cell lung cancer (SCLC) and non-small cell lung cancer (NSCLC). SCLC is notable due to its highly aggressive and lethal clinical course, defined by rapid tumor growth, early dissemination, and metastasis (*Gazdar et al., 2017*). SCLC is a neuroendocrine (NE) tumor (*Travis et al., 2015*), and recent studies have demonstrated that it is a molecularly heterogeneous disease comprising discrete tumor subtypes defined by expression of different transcriptional regulators, namely achaete-scute homolog 1 (ASCL1) and neurogenic differentiation factor 1 (NEUROD1), which together account for approximately 80% of SCLC cases (*Borromeo et al., 2016*; *Rudin et al., 2019*). ASCL1 and NEUROD1, along with insulinoma-associated protein 1 (INSM1) and POU class 3 homeobox 2 (BRN2), are recognized as important master regulators for NE differentiation in SCLC (*Rudin et al., 2019*; *Fujino et al., 2015*; *Ishii et al., 2013*). Besides NE differentiation, SCLC is

further distinguished from other major NSCLC subtypes such as lung adenocarcinoma (LUAD) and squamous cell carcinoma by its unique cellular morphology (*Rudin et al., 2019*) and genetic hallmarks including frequent inactivation of tumor suppressors *TP53* and *RB1* (*Peifer et al., 2012*; *George et al., 2015*). SCLC is also characterized by the absence of EGFR expression (*Gamou et al., 1987*) and low activity of the downstream mitogen-activated protein kinase (MAPK) pathway (*Byers et al., 2012*). Furthermore, activating alterations in *EGFR* and *KRAS*, which are highly prevalent in LUAD (*Cancer Genome Atlas Research Network, 2014*), are rarely identified in SCLC (*Peifer et al., 2012*; *George et al., 2015*) (Summarized in *Figure 1a*). Despite developing in the same organ and having exposure to the same etiological agent in most instances, no biological rationale aside from cell of origin has been provided to explain these divergent molecular characteristics. Therefore, elucidating the factors that underlie the selection of specific genetic drivers in different lineage contexts may yield insights toward the development and progression of these lung cancer types.

In contrast to SCLC, for which no major treatment breakthroughs have been made in the last two decades, LUAD treatment has greatly benefitted from targeted therapies for driver oncogenes, highlighted by the success of those inhibiting *EGFR*-mutant tumors (*Maemondo et al., 2010*; *Soria et al., 2018*). However, resistance to molecular targeted therapy is inevitable and long-term cures remain elusive. Histological transformation from LUAD to SCLC (*Memorial Sloan-Kettering Cancer Center Lung Cancer OncoGenome Group et al., 2006*) occurs in 5–15% of cases with acquired resistance to EGFR tyrosine kinase inhibitors (TKIs) (*Sequist et al., 2011*; *Westover et al., 2018*), typically after a long duration (median ≥13 months) of TKI treatment (*Ferrer et al., 2019*; *Offin et al., 2019*). This lineage transition may become a more prominent and important resistance mechanism in the future with the approval of the third-generation EGFR-TKI osimertinib (*Leonetti et al., 2019*) as a first-line therapy, as this drug has better on-target inhibition and overcomes the most common resistance mechanism to earlier generation EGFR-TKIs, the T790M mutation (*Mok et al., 2017*), and provides longer progression-free survival (*Soria et al., 2018*). *EGFR*-mutant LUADs undergoing TKI treatment are known to be at unique risk of histological transformation to SCLC (*Ferrer et al., 2019*) particularly when p53 and RB are concurrently inactivated (*Offin et al., 2019*; *Niederst et al., 2015*; *Lee et al., 2017*). Surprisingly, *EGFR*-mutant tumors lose EGFR protein expression (*Niederst et al., 2015*) after small cell transformation, mimicking de novo SCLC, despite retaining the initial activating mutation in *EGFR* (*Ferrer et al., 2019*; *Niederst et al., 2015*). Furthermore, TKI-resistant *EGFR*-mutant LUADs that have undergone SCLC transformation typically lack the acquisition of other genetic alterations associated with TKI resistance that are known to reactivate MAPK signaling (*Roper et al., 2020*). However, the biological mechanisms regulating the SCLC transformation process remain unknown, because no in vitro or in vivo model systems have been established to date that enable the comprehensive study of this phenomenon.

Based on the above observations, and that SCLC-transformed LUAD resembles de novo SCLC in terms of molecular features, we hypothesized that there is a unique interplay between MAPK signaling and suppression of NE differentiation in lung cancer. Further, we anticipated that understanding this interplay would reveal the factors that underpin the selection of specific genetic alterations in the development of the different lung cancer subtypes and acquisition of drug resistance. Therefore, we aimed to investigate the consequences of LUAD oncogene expression and activation of MAPK pathway signaling in SCLC cells in order to potentially provide mechanistic insight into the programs driving small cell lineage transformation in the context of EGFR-TKI resistance.

## Results

### Mutually exclusive association between MAPK activation and NE marker expression in lung cancer

Previous studies have demonstrated that SCLC and LUAD differ in their expression and activation of MAPK signaling components, and that SCLC-transformed LUAD loses EGFR expression. To first assess the relationship between EGFR status and NE marker expression in lung cancer, we performed western blot analysis across a diverse panel of lung cancer cell lines. Lysates from eight *EGFR*-mutant, four *KRAS*-mutant, and two *EGFR/KRAS* wild-type LUAD, as well as two large cell carcinoma and four SCLC cell lines were assessed (*Figure 1—figure supplement 1a*). All the SCLC cell

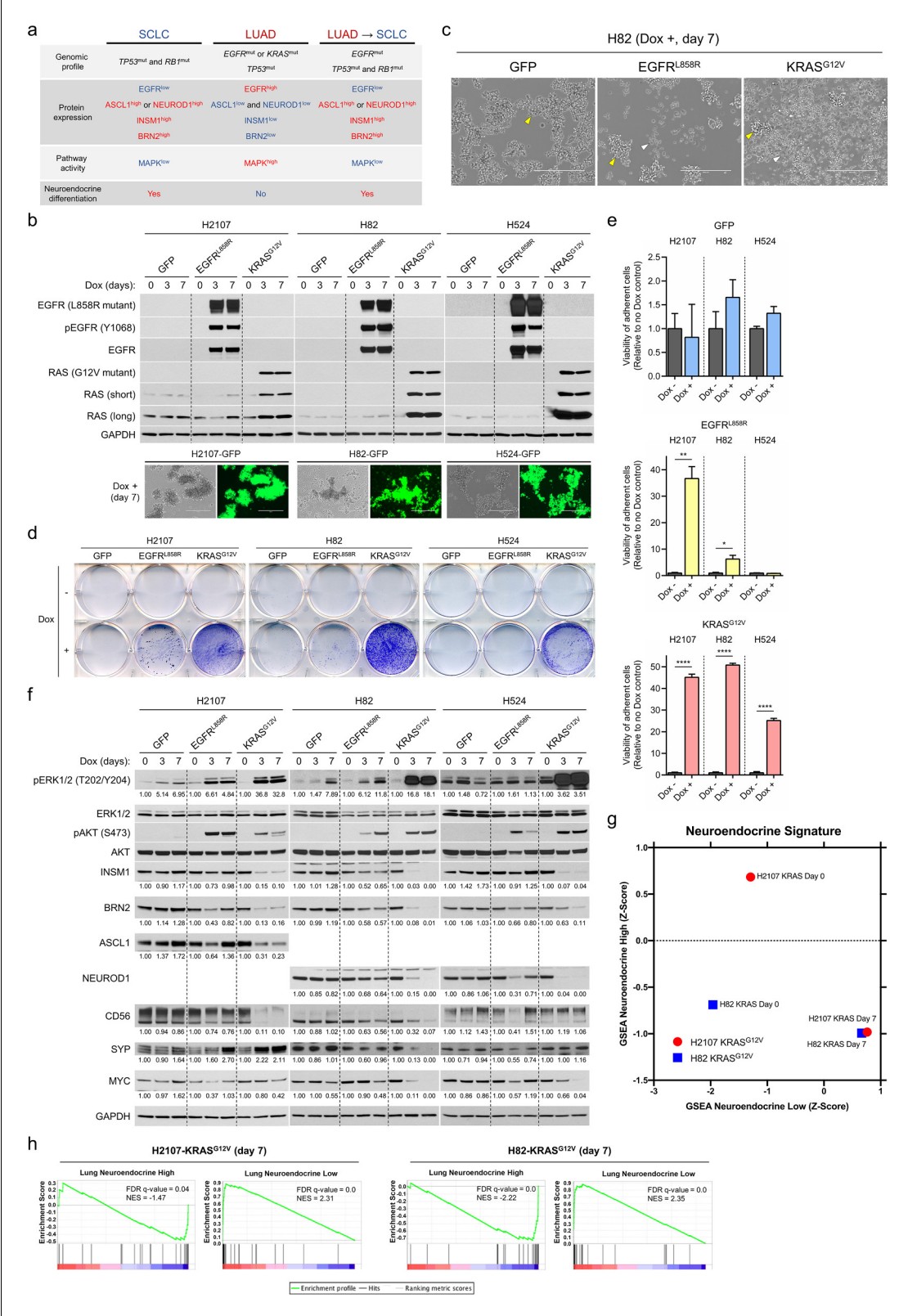

**Figure 1.** Effects of mutant KRAS or EGFR expression on phenotype and neuroendocrine markers in small cell lung cancer cells. (**a**) Overview of representative somatic alterations, protein expression profiles, MAPK pathway activity, and neuroendocrine differentiation in small cell lung cancer (SCLC), lung adenocarcinoma (LUAD), and transformed SCLC from *EGFR*-mutated LUAD. (**b**) Induction of EGFR^L858R or KRAS^G12V as assessed by western blot and of GFP assessed by fluorescence phase contrast images in SCLC cell lines, H2107, H82, and H524 cells, upon treatment with 100 ng/

*Figure 1 continued on next page*

*Figure 1 continued*

mL doxycycline (dox) for 3 and 7 days. GAPDH was used as a loading control. (**c**) Photomicrographs showing the growing morphology in suspending aggregates of GFP-overexpressing H82 cells (left) and in mixed adherent and suspended states of EGFR^L858R- (middle) or KRAS^G12V- (right) overexpressing H82 cells upon treatment with 100 ng/mL dox for 7 days. Yellow and white arrowheads indicate suspending aggregates and adherent cells, respectively. Scale bars, 400 μm. (**d**) Crystal violet assay of adherent cells with or without induction of GFP, EGFR^L858R, or KRAS^G12V in H2107 (on day 7), H82 (on day 7), and H524 (on day 5) cells. Medium containing suspended cells was removed, and adherent cells were washed with PBS and then fixed and stained with crystal violet. (**e**) Quantification of cell attachment after GFP, EGFR^L858R, or KRAS^G12V induction in H2107 (on day 7), H82 (on day 7), and H524 (on day 5) cells. After incubation, medium containing suspended cells was removed and adherent cells were washed with PBS. Adherent cells were then cultured in fresh media and viability was assessed using an alamarBlue cell viability agent. Values relative to a no dox control for each cell line are graphed as mean (three biological replicates) ± SEM. The Student's *t* test, ****p<0.0001; **p<0.01; and *p<0.05. (**f**) Representative immunoblot showing the effects of induction of GFP, EGFR^L858R, or KRAS^G12V on neuroendocrine markers as well as phosphorylation status of ERK and AKT upon treatment with 100 ng/mL dox for 3 and 7 days in H2107, H82, and H524 cells. GAPDH was used as a loading control. Numbers below blots show the amounts of respective band relative to the corresponding non-dox-treated control values after normalization to ERK1/2 (for pERK1/2) or GAPDH (for INSM1, BRN2, ASCL1, NEUROD1, CD56, SYP, and MYC) in each condition for each cell line. (**g**) Gene set enrichment analysis (GSEA) neuroendocrine differentiation scores of H2107-KRAS^G12V and H82-KRAS^G12V cells pre- and post-dox treatment (day 7). (**h**) Enrichment plots of the 50-gene lung-cancer-specific neuroendocrine expression signature gene sets in H2107-KRAS^G12V and H82-KRAS^G12V cells post-dox treatment for 7 days compared with GFP-overexpressing controls. Immunoblots are representative of at least two biological replicates.

The online version of this article includes the following figure supplement(s) for figure 1:

**Figure supplement 1.** Profiling of EGFR and neuroendocrine factor expression and genomic alterations in the *EGFR* and *KRAS* genes in lung cancer.
**Figure supplement 2.** Effects of EGFR^L858R or KRAS^G12V overexpression in small cell lung cancer cell lines.
**Figure supplement 3.** Assessment of heterogeneity in KRAS^G12V-transduced small cell lung cancer cell lines.

lines completely lacked EGFR protein expression and a large cell carcinoma with NE differentiation cell line, H1155, weakly expressed EGFR, whereas all other cell lines universally expressed high levels of EGFR. Inversely, expression of four NE transcription factors (TFs) – INSM1, BRN2, ASCL1, and NEUROD1 – as well as NE markers CD56 and synaptophysin (SYP) were largely specific to SCLC cell lines. Thus, there was a clear inverse association between NE differentiation and EGFR expression in lung cancer. We also observed a mutually exclusive expression pattern between ASCL1 and NEUROD1 in the five NE cell lines, whereas INSM1 and BRN2 were broadly expressed in these cell lines (*Figure 1—figure supplement 1a*). We next explored mutation and copy number alteration status of *KRAS* and *EGFR* using publicly available whole-genome sequencing, whole-exome sequencing, and The Cancer Genome Atlas (TCGA) data sets. As reported (*Peifer et al., 2012*; *George et al., 2015*), the prevalence of genomic alterations in these two oncogenes in SCLC was low. In particular, genomic alterations in the *KRAS* gene were never identified in SCLC (*Figure 1—figure supplement 1b*).

## Expression of mutant KRAS or EGFR in SCLC induces trans-differentiation into a NSCLC-like state with suppressed NE differentiation

To determine whether the inverse association between MAPK activity and NE differentiation is due to differences in cell of origin for the specific cancer types or instead attributed to the direct signaling pathways regulated by the mutated oncogenes, we conditionally expressed either EGFR^L858R or KRAS^G12V, which are the most prevalent drivers in LUAD (*Cancer Genome Atlas Research Network, 2014*), as well as a GFP control in three SCLC cell lines; H2107 (ASCL1-high; SCLC-A *Rudin et al., 2019*), H82 (NEUROD1-high; SCLC-N *Rudin et al., 2019*), and H524 (SCLC-N). Western blots confirmed successful induction of these oncoproteins under the tight control of an inducible TetO promoter using doxycycline (*Figure 1b*). Consistent with the results shown in previous reports (*Falco et al., 1990*; *Calbo et al., 2011*) in which HRAS or RAS^V12 were retrovirally transduced in SCLC cell lines, the SCLC-N cell lines both demonstrated a phenotypic transition from a suspended to adherent state after KRAS^G12V induction, with the most striking change occurring in H82 cells (*Figure 1c–e*). However, in contrast to a previous study where the cell lines representing the classic subtype of SCLC, which are currently classified as SCLC-A (*Rudin et al., 2019*), showed no discernible phenotypic changes in response to HRAS expression (*Mabry et al., 1988*), we found that H2107 SCLC-A cells also demonstrated this phenotypic transition (*Figure 1d and e*). While EGFR^L858R transduction also induced a shift to an adherent state in H2107 and H82 cells, this growth pattern was

more modest than that observed with KRAS$^{G12V}$ (*Figure 1d and e*). Furthermore, the phenotypic effect of EGFR$^{L858R}$ expression was temporally delayed compared to KRAS$^{G12V}$ with cells forming suspension clusters first, then subsequently migrating to become adherent, whereas KRAS$^{G12V}$ induced direct formation of adherent cells that were diffusely distributed (*Figure 1d* and *Figure 1— figure supplement 2a*). The impact of oncogene induction on cell viability was also assessed, and we observed variable effects across the three cell lines (*Figure 1—figure supplement 2b and c*). Concordant with the positive effect of KRAS$^{G12V}$ on cell viability in H82 cells, forced expression of KRAS$^{G12V}$ increased anchorage-independent growth in soft agar (*Figure 1—figure supplement 2d*). There was a clear increase in cleaved PARP after doxycycline treatment in H2107 cells (both EGFR$^{L858R}$ and KRAS$^{G12V}$) as well as in H524 cells (EGFR$^{L858R}$), suggesting that cell death, at least partially, accounts for the decreased cellular viability in these cells (*Figure 1—figure supplement 2e*).

Given that established SCLC cell lines typically grow in suspension as aggregated cells (*Gazdar et al., 1980*) and the NE type H1155 large cell carcinoma cells also grow partly in suspension clusters, we considered that there might be a relationship between a suspended cellular growing phenotype and NE differentiation. Thus, shift in cell growth patterns of SCLC cells from suspension to an adherent state after induction of LUAD mutant oncogene expression suggested that oncogenic signaling may lead to lineage transformation in SCLC. To further assess this, we determined the impact of induced mutant EGFR or KRAS on the expression of the main neuroendocrine transcription factors (NETFs) - including INSM1, BRN2, ASCL1, and NEUROD1 – in the SCLC cell lines. The four NETFs were all downregulated by the oncoproteins, which was again more prominently observed with KRAS$^{G12V}$ expression than with EGFR$^{L858R}$ (*Figure 1f*). To globally assess lineage status, we profiled the transcriptional changes in H2107 and H82 cells following doxycycline treatment to induce EGFR$^{L858R}$ or KRAS$^{G12V}$ for both acute (24 hr) and long-term (7 days) durations and compared to respective GFP controls. Gene set enrichment analysis (GSEA) using a 50-gene lung cancer-specific NE expression signature (*Zhang et al., 2018*) revealed a shift from high NE differentiation at baseline to low NE differentiation after induction of KRAS$^{G12V}$ at the day seven time point, with genes associated with NE status becoming downregulated and those typically low in SCLC demonstrating high expression (*Figure 1g and h*). Importantly, extracellular signal-regulated kinases (ERK1 and ERK2) were more strongly phosphorylated by KRAS$^{G12V}$ than EGFR$^{L858R}$ (*Figure 1f*), suggesting a potential rationale for the differential effects of induction of these oncoproteins on phenotype and NE marker expression. Based on these results, we used the KRAS$^{G12V}$ transduction model for further experiments.

Recent evidence has demonstrated that MYC can dynamically drive a shift of master NETFs of SCLC from ASCL1 to NEUROD1 to YAP1 in the context of RB and p53 loss (*Ireland et al., 2020*). We found that MYC protein levels were downregulated by EGFR$^{L858R}$ and KRAS$^{G12V}$ (*Figure 1f*) despite previous reports that ERK-mediated phosphorylation of MYC prevents MYC degradation (*Sears et al., 2000*). Furthermore, GSEA indicated downregulation of an MYC target gene set in H82-KRAS$^{G12V}$ cells compared with GFP control cells on day 7 of doxycycline treatment (*Figure 1— figure supplement 2f*). To further determine whether the downregulation of NETFs after oncogene induction is in the context of MYC-driven master NETF shift, we next evaluated YAP1 expression in SCLC cell lines after KRAS$^{G12V}$ transduction, as YAP1 is a marker of non-NE SCLC (*Rudin et al., 2019*). Despite all three SCLC cell lines demonstrating suppression of NETFs after KRAS$^{G12V}$ induction, *YAP1* was significantly upregulated only in H82-KRAS$^{G12V}$ cells treated with doxycycline for 7 days (3.3-fold, *Supplementary file 1*), which was mirrored by the weakly detectable YAP1 protein level in the same cell line by western blot (*Figure 1—figure supplement 2g*). These results suggest that the mutant EGFR- and KRAS-induced shift from a high to low NE phenotype in SCLC cell lines is unlikely to be a subclass transition driven by a MYC-YAP1 axis.

We noted that mutant EGFR or KRAS-induced SCLC cell lines showed a mixed phenotype comprising both suspended and adherent cells after doxycycline treatment. Thus, we asked whether this heterogeneity in growth pattern was derived from the polyclonal nature of transduced cells. To address this, we established single cell-derived clones and found that clonal cells also showed a mixture of adherent and suspended cells after KRAS$^{G12V}$ induction (*Figure 1—figure supplement 3a*). We also profiled the expression status of the NE factors in adherent, suspended, or mixed populations, separately, in the subacute (doxycycline day 3) and chronic (doxycycline day 28) phases after KRAS$^{G12V}$ induction using polyclonal cells (*Figure 1—figure supplement 3b*). Despite relatively

similar induced levels of KRAS$^{G12V}$ as well as phospho-ERK1/2 in the subacute phase, adherent cells lost NE factors to a greater degree than suspended cells. In terms of the growth state, isolated adherent cells gave rise to both adherent and suspended cells after serial passages under doxycycline treatment; however, the proportion of adherent cells became lower with each passage, with a dramatic reduction observed after 2 weeks. Nonetheless, NE markers were still suppressed in the remaining adherent cell population. Importantly, isolated suspended cells did not give rise to adherent cells after serial passages and KRAS$^{G12V}$ expression was highly attenuated in this subset of cells, even in H82 cells in which forced expression of mutant-KRAS exhibited an advantageous effect on cell proliferation. This suggests negative selection of KRAS$^{G12V}$-positive cells or epigenetic silencing of transduced *KRAS$^{G12V}$* in the long-term (28 day) culture driven by the incompatibility of *KRAS* activation in SCLC biology. Together, these data suggest that constitutive activation of MAPK pathway by mutant KRAS and EGFR affects the growth phenotype and suppresses NE differentiation program in SCLC in a heterogenous manner.

## ERK activation inhibits expression of NETFs in SCLC

ERK is the central pathway node of MAPK signaling and acts to phosphorylate hundreds of downstream targets and control many fundamental cellular processes (*Yoon and Seger, 2006*). Thus, we hypothesized that ERK may be the main mediator of the multiple effects observed in SCLC cells after mutant EGFR or KRAS induction. This was suggested by the differential effects of mutant EGFR versus mutant KRAS transduction in SCLC cells, where the latter induced more prominent changes and was associated with increased levels of phospho-ERK1/2 (*Figure 1f*). We tested this by treating TetO-KRAS$^{G12V}$-transduced SCLC cells with an ERK1/2 inhibitor, SCH772984, and found that this compound rescued the suppression of NETFs after doxycycline induction (*Figure 2a*). To confirm that this rescue was not attributed to off-target effects of SCH772984, we also performed genetic knockdown of either *ERK1* (*MAPK3*), *ERK2* (*MAPK1*), or both. As shown in *Figure 2b*, expression of NE factors was restored by transfection of siRNAs targeting *ERK2* but not *ERK1*, indicating that ERK2 is a dominant node mediating this process. *ERK1* knockdown likely augmented ERK2 activity by disruption of negative feedback signaling as previously described (*Unni et al., 2018*), and therefore did not restore the repressed NE factors when inhibited alone.

In addition to MAPK, the phosphoinositide 3-kinase (PI3K)/AKT pathway is another major signaling arm-activated downstream of EGFR and RAS. Indeed, phospho-AKT levels were increased after oncogenic EGFR or KRAS induction in our model (*Figure 1f*). To test whether this pathway was also involved in suppression of NE factors in SCLC cells after oncogene induction, we treated KRAS$^{G12V}$-induced cells with an AKT-inhibitor, MK-2206. Despite near complete suppression of phosphorylated AKT with MK-2206, decreased NETF expression was still observed upon doxycycline treatment, suggesting that the PI3K/AKT pathway was not responsible for the NE dedifferentiation effects observed (*Figure 2c*).

## ERK in combination with AKT activation drives phenotypic growth state change in SCLC after Oncogene induction

To assess mediators of the attached growth phenotype, we quantified cellular state and viability of KRAS$^{G12V}$-induced SCLC cells with or without SCH772984, MK-2206, or combination of these drugs. To minimize the bias from potential toxicities of KRAS$^{G12V}$ induction combined with drug treatments, we used an acute incubation time of 72 hr. In H82 and H524 cells, the combined inhibition of ERK and AKT reversed the suspended-to-adherent phenotypic transition which was not seen with either ERK or AKT inhibition alone (*Figure 2—figure supplement 1a and b*). To exclude the possibility that applied drugs might have a lethal impact on cells, resulting in less adhesion in a non-specific manner, we also assessed the viability of the whole cell population after the combination drug treatment and observed no obvious adverse effects (*Figure 2—figure supplement 1c*). In contrast to H82 and H524, KRAS$^{G12V}$-induced cell attachment was significantly enhanced by ERK inhibition in H2107 cells, which was not completely rescued by additional AKT inhibition (*Figure 2—figure supplement 1a and b*).

Lastly, we aimed to determine potential downstream effectors of ERK that are responsible for mediating the cellular phenotypic state change in conjunction with AKT after mutant KRAS induction in SCLC. We assessed mitogen- and stress-activated protein kinase (MSK)/ribosomal S6 kinase (RSK)

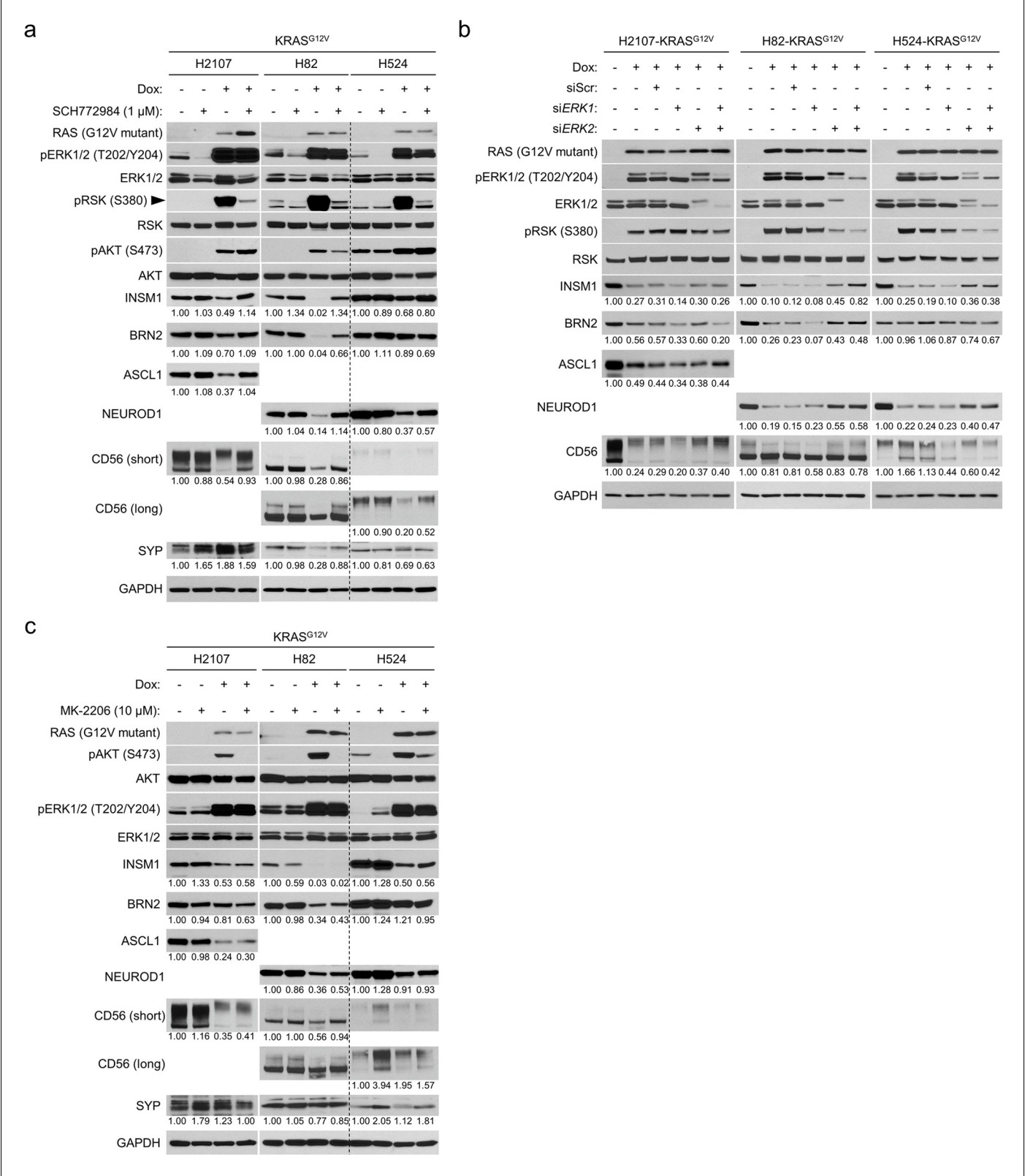

**Figure 2.** Hyperactivated ERK represses expression of neuroendocrine transcription factors in small cell lung cancer. (a) Western blot showing the ERK inhibitor (SCH772984 [1 μM])-mediated restoration of neuroendocrine transcription factors that are repressed by KRAS^G12V induction in small cell lung cancer cell lines H2107, H82, and H524. Cells were treated with indicated agents for 72 hr. GAPDH was used as a loading control. Numbers below blots show the amounts of each band relative to the corresponding non-doxycycline (dox)-treated and non-SCH772984-treated (DMSO-treated) control

*Figure 2 continued on next page*

Figure 2 continued

values (set to one in each cell line panel) after normalization to GAPDH. (b) Western blot demonstrating the effects of KRAS[G12V] induction and treatment with siRNA pools targeting *ERK1*, *ERK2*, or both on expression of neuroendocrine transcription factors in the small cell lung cancer cell lines. Cells were treated with 100 ng/mL dox and indicated siRNAs for 72 hr. Scrambled siRNA (siScr) was used as a negative control. GAPDH was used as a loading control. Numbers below blots show the amounts of each band relative to the corresponding non-dox-treated and non-siRNA-treated control values (set to one in each cell line panel) after normalization to GAPDH. (c) Western blot showing the effects of AKT inhibition using MK-2206 (10 μM) on expression of neuroendocrine factors that are suppressed by KRAS[G12V] in the small cell lung cancer cell lines. Cells were treated with indicated agents for 72 hr. GAPDH was used as a loading control. Numbers below blots show the amounts of each band relative to the corresponding non-dox-treated and non-MK-2206-treated (DMSO-treated) control values (set to one in each cell line panel) after normalization to GAPDH. Immunoblots are representative of at least two biological replicates.

The online version of this article includes the following figure supplement(s) for figure 2:

**Figure supplement 1.** Effects of ERK and/or AKT inhibition on the KRAS[G12V]-mediated phenotypic change in small cell lung cancer.

**Figure supplement 2.** Effects of MSK/RSK and/or AKT inhibition on the KRAS[G12V]-mediated phenotypic change in small cell lung cancer.

**Figure supplement 3.** Profiling of phospho-kinases with or without MSK/RSK inhibition in H82- and H2107-KRAS[G12V] cells.

**Figure supplement 4.** HES1 is induced by ERK independently from NOTCH signaling but does not suppress neuroendocrine transcription factors in small cell lung cancer cell lines.

**Figure supplement 5.** Upregulation of REST and SOX9 by ERK is not responsible for the suppressed neuroendocrine differentiation in KRAS[G12V]-transduced small cell lung cancer cell lines.

for this purpose as they are direct downstream effectors of ERK1/2 and inhibited these alone or in combination with AKT after doxycycline induction. Interestingly, cell attachment was not reversed by the combined MSK/RSK and AKT inhibition but was instead enhanced in the context of MSK/RSK suppression, particularly in H2107 cells (*Figure 2—figure supplement 2a–c*). We conducted phospho-kinase profiling with or without MSK/RSK inhibition using H82- and H2107-KRAS[G12V] cells under doxycycline treatment and this revealed that phospho-AKT as well as phospho-ERK1/2 levels were increased after MSK/RSK inhibition, particularly in H2107-KRAS[G12V] cells (*Figure 2—figure supplement 3*). This feedback activation explains why MSK/RSK inhibition did not rescue the phenotypic change after KRAS[G12V] induction and suggests that other ERK effectors mediate these effects in conjunction with AKT. Together, these results suggest that the activation of both ERK and AKT is required for the transition of the growth phenotype in SCLC, while ERK2 is a central hub of the oncogene-induced suppression of NE regulators.

## NOTCH signaling is activated by ERK upon KRAS induction in SCLC but is not responsible for repression of NE factors

To examine the mechanisms of ERK-mediated suppression of NETFs in SCLC, we identified differentially expressed genes between EGFR[L858R] vs GFP and KRAS[G12V] vs GFP cells at each time point for both H82 and H2107 with and without doxycycline (*Supplementary file 1*). As summarized in *Figure 2—figure supplement 4a*, the overlap between the two cell lines following KRAS[G12V] induction included 65 and 381 upregulated (>1.5 fold) and 3 and 70 downregulated (<0.67 fold) genes on day 1 and day 7, respectively. Mirroring the differential activation of ERK and NE suppression by the oncogenes, there were fewer genes differentially expressed in the cell lines upon EGFR[L858R] induction (*Figure 2—figure supplement 4b*), and therefore, we focused on the KRAS[G12V] model system to identify candidates. Among the commonly upregulated genes in the two cell lines after 7 days of KRAS[G12V] induction, hairy and enhancer of split 1 (*HES1*) was one of the top differentially expressed genes in H2107 cells. *HES1* was of interest as a candidate gene suppressing NE differentiation in our model as it functions as a critical transcriptional repressor of neuronal differentiation under control of NOTCH signaling (*Iso et al., 2003*). Furthermore, decreased *HES1* expression was recently shown to be associated with NE differentiation upon osimertinib resistance in *EGFR*-mutant LUAD patient samples (*Roper et al., 2020*). Immunoblots validated the strong induction of HES1 protein in H2107 and H524 cells by mutant EGFR and KRAS, while the activated form of NOTCH1, cleaved NOTCH1, was paradoxically decreased (*Figure 2—figure supplement 4c*). Further, HES1 was weakly induced without the presence of cleaved NOTCH1 in H82 cells, which harbor a *NOTCH1* missense mutation (V776M), while induction of HES1 was completely suppressed by pharmacological ERK inhibition (*Figure 2—figure supplement 4d*). We then tested whether blockade of NOTCH signaling by a γ-secretase inhibitor, RO4929097, prevents HES1 induction by KRAS[G12V] and found no effect in H82

and H524 cells, although it was partially attenuated in H2107 cells (*Figure 2—figure supplement 4e*). Likewise, suppressed NETFs were not rescued by this treatment. We next carried out *HES1* knockout in KRAS$^{G12V}$-inducible cells, but elimination of HES1 did not restore NE factors suppressed by activated ERK (*Figure 2—figure supplement 4f*). These data suggest that oncogene-mediated ERK activation in SCLC induces HES1 independently from NOTCH signaling; however, induced HES1 does not underlie the ERK-mediated suppression of NETFs.

## SOX9 and REST transcription programs are mediated by mutant KRAS induction in SCLC cells

As HES1 upregulation was not responsible for the suppression of NE differentiation, we next assessed whether the differentially expressed genes in SCLC after mutant KRAS induction were enriched for specific transcriptional programs that could indicate a potential mediator of this effect. We identified enrichment for targets regulated by RE1-silencing TF (REST) and SRY-related high-mobility group box 2 (SOX2) in both H82 and H2107 cells after mutant KRAS induction (*Figure 2—figure supplement 5a*). REST is a transcriptional repressor of neuronal genes and is a direct target of NOTCH1 (*Lim et al., 2017*), making it a logical candidate for repressing NE factors under control of activated ERK in our system. In fact, in addition to its downstream targets, microarray data also showed upregulation of *REST* itself by KRAS$^{G12V}$ in H2107 and H82 cells (*Figure 2—figure supplement 5b*), which was validated by RT-qPCR (*Figure 2—figure supplement 5c*). As opposed to a previous study (*Lim et al., 2017*), however, introduction of *REST* siRNAs – while effective at knocking down *REST* levels – did not contribute to restoration of ERK-mediated suppression of NE factors (*Figure 2—figure supplement 5c*).

The SOX family TFs are potent drivers of direct somatic cell reprogramming into multiple lineages (*Julian et al., 2017*). We reasoned that SOX9 but not SOX2 might be a candidate TF to explain the lineage transition in our model, because SOX2 was expressed in only H2107 cells both before and after doxycycline treatment (*Figure 2—figure supplement 5d*), while SOX9 expression has been reported to negatively associate with SOX2 expression (*Lin et al., 2016*). In addition, distal lung cells including alveolar epithelial type 2 cells are identified by SOX9 expression (*Laughney et al., 2020*), and SOX9 was shown to associate with POU class 2 homeobox 3 (POU2F3)-driven subtype of SCLC (*Huang et al., 2018*), which represents a subtype of SCLC lacking typical NE markers (*Rudin et al., 2019*). Furthermore, a recent study demonstrated that *SOX9* expression is enriched in ASCL1-low human SCLC cell lines and that SOX9 target genes are enriched in ASCL1-low human SCLC tumors (*Olsen, 2020*). Despite *SOX9* transcript being upregulated by KRAS$^{G12V}$ only in H2107 cells in the microarray data, we found that SOX9 protein was upregulated by mutant KRAS in all the three cell lines (*Figure 2—figure supplement 5d*), and this was prevented by ERK inhibition (*Figure 2—figure supplement 5e*). However, CRISPR/Cas9-mediated *SOX9* knockout demonstrated no effects on expression levels of NE factors after KRAS$^{G12V}$ induction (*Figure 2—figure supplement 5f*). Together, these data suggest that while ERK signaling induces expression of HES1, REST and SOX9, these TFs are not responsible for the lineage transformation observed after LUAD oncogene induction in SCLC.

## ERK activation in SCLC induces global chromatin modifications

We next investigated whether ERK causes chromatin remodeling in SCLC that could explain the mechanisms by which NETFs are suppressed by constitutive activation of ERK. Indeed, global levels of histone marks – which can be used to classify enhancers – were revealed to be altered after EGFR$^{L858R}$ or KRAS$^{G12V}$ induction in SCLC cells (*Figure 3*). Specifically, these oncoproteins dramatically increased the active enhancer marks histone 3 lysine 9 acetylation (H3K9ac), H3K14ac, and H3K27ac in H82 and H524 cells, whereas they decreased histone 3 lysine 4 tri-methylation (H3K4me3) in H2107 cells. These data suggest that hyperactivated ERK-mediated suppression of NE factors in SCLC might be dependent on altered chromatin structures, which vary depending on the subtype of SCLC defined based on the corresponding master regulator, ASCL1 or NEUROD1.

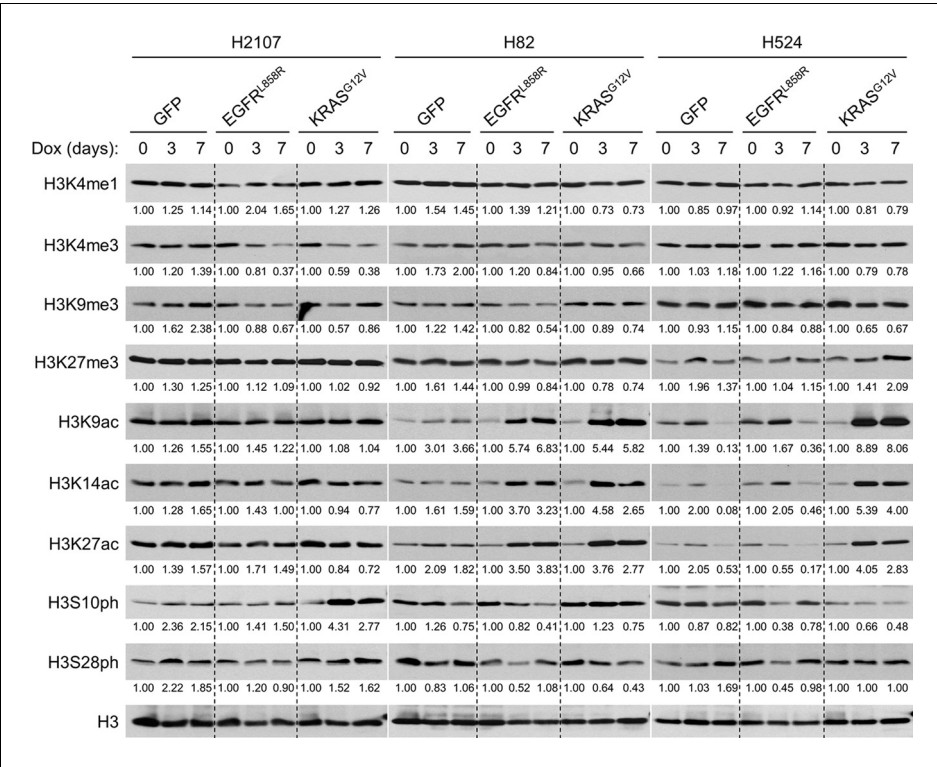

**Figure 3.** Oncogene-mediated ERK activation alters the global chromatin modifications characterized by histone marks. Western blot showing the effects of transduction of GFP, EGFR[L858R], or KRAS[G12V] on histone marks upon treatment with 100 ng/mL doxycycline (dox) for 3 and 7 days in H2107, H82, and H524 cells. H3 was used as a loading control. Numbers below blots show the amounts of each band relative to the corresponding non-dox-treated control values after normalization to H3 in each condition for each cell line. Immunoblots are representative of at least two biological replicates.

## ERK activation suppresses NETFs through reorganization of active chromatin

Prominently increased H3K27ac after mutant oncogene induction in SCLC was of interest as over 90% of H3K27ac in cells is dependent on two histone acetyltransferases (HATs) – cAMP-response-element-binding protein (CREB)-binding protein (CBP)/*CREBBP* and its homologous p300/*EP300* (*Jin et al., 2011*) – which are recurrently inactivated by mutations in SCLC (*Peifer et al., 2012*; *George et al., 2015*; *Rudin et al., 2012*). In addition, a clonal evolution study showed an *EP300* rearrangement in an *EGFR*-mutant tumor before transforming to SCLC through EGFR-TKI treatment (*Lee et al., 2017*). *CREBBP* mutations were also shown to be enriched in *EGFR*-mutant LUAD tumors that subsequently underwent TKI-induced SCLC transformation (*Offin et al., 2019*). Reciprocally, ERK1 and ERK2 are known to directly phosphorylate and activate CBP (*Ait-Si-Ali et al., 1999*) and p300 (*Liu et al., 2016*), respectively. ERK also indirectly activates HAT activity of CBP/p300 through phosphorylation of MSK1/2, which results in phosphorylation of histone 3 serine 28 (*Soloaga et al., 2003*) and recruitment and activation of CBP/p300 (*Josefowicz et al., 2016*; *Figure 4a*). Together, this suggests that SCLC tumors evolve in a manner that selects for decreased H3K27ac levels to maintain their NE phenotype, and that activation of CBP and p300 by ERK may lead to lineage transformation.

To clarify the dependency on MSK1/2 in the regulation of NE factors by ERK, we treated KRAS[G12V]-inducible cells with or without doxycycline and a compound (SB-747651A) that inhibits MSK as well as RSK (*Naqvi et al., 2012*). As shown in *Figure 4b*, phosphorylation of CREB, a downstream target of MSK1/2, was well inhibited by this compound and phospho-AKT levels were again upregulated in H2107 and H524 cells as shown in *Figure 2—figure supplement 3*. MSK inhibition modestly prevented the suppression of BRN2 and NEUROD1, but INSM1 was not rescued in H82

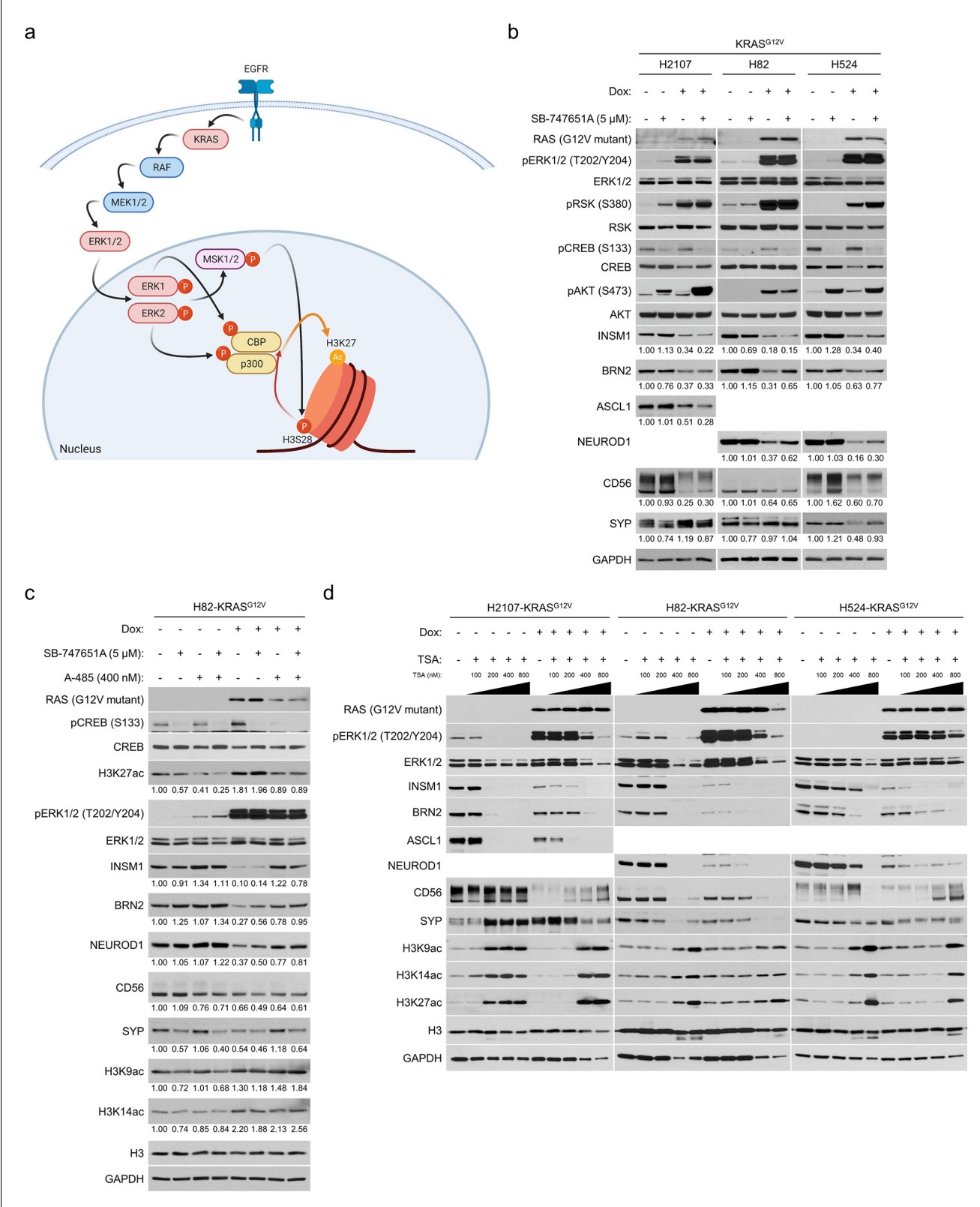

**Figure 4.** ERK-mediated histone 3 lysine 27 acetylation (H3K27ac) is responsible for suppression of neuroendocrine transcription factors in small cell lung cancer. (a) Known model of receptor tyrosine kinase/RAS/ERK pathway-mediated promotion of H3K27ac. Illustration was created with BioRendrer. com. (b) Western blot showing the effects of MSK/RSK inhibition using 5 μM SB-747651A on expression of neuroendocrine factors as well as on ERK/ RSK/CREB pathway activity and AKT phosphorylation with or without KRAS$^{G12V}$ transduction for 72 hr. GAPDH was used as a loading control. Numbers

*Figure 4 continued on next page*

Figure 4 continued

below blots show the amounts of each band relative to the corresponding non-doxycycline (dox)-treated and non-SB-747651A-treated (DMSO-treated) control values (set to one in each cell line panel) after normalization to GAPDH. (c) Western blot of neuroendocrine markers and histone 3 lysine acetylation marks in H82-KRAS$^{G12V}$ cells. The cells were treated with SB-747651A (5 μM), a CBP/p300 inhibitior A-485 (400 nM), or both, as well as 100 ng/mL dox for 72 hr. H3 and GAPDH were used as loading controls. Numbers below blots show the amounts of each band relative to the corresponding non-dox-treated and non-drug-treated (DMSO-treated) control values (set to lane 1) after normalization to H3 (for H3K27ac, H3K9ac, and H3K14ac) or GAPDH (for INSM1, BRN2, NEUROD1, CD56, and SYP). (d) Western blot of neuroendocrine markers after inhibition of histone deacetylases using trichostatin A (TSA) at different concentrations with or without KRAS$^{G12V}$ transduction for 72 hr. H3 and GAPDH were used as loading controls. Immunoblots are representative of at least two biological replicates.

The online version of this article includes the following figure supplement(s) for figure 4:

**Figure supplement 1.** Inhibition of CBP/p300 restores neuroendocrine transcription factors suppressed by ERK in H82-KRAS$^{G12V}$ cells.

and H524 cells. Furthermore, no effects were observed in H2107 cells by this treatment. We next inhibited CBP/p300 in KRAS$^{G12V}$-inducible cells using A-485, a potent and selective inhibitor of the catalytic function of CBP/p300 (*Lasko et al., 2017*), and revealed that at an optimized concentration (400 nM), A-485 restored INSM1, BRN2, and NEUROD1 expression and reduced H3K27ac to a basal level in H82 cells after mutant KRAS induction (*Figure 4c* and *Figure 4—figure supplement 1a*). Treatment with A-485 did not affect the levels of two other histone three lysine acetylation marks, H3K9ac and H3K14ac, and inhibition of MSK/RSK did not consistently show additive rescue effects for NE markers (*Figure 4c*). We also treated H82-KRAS$^{G12V}$ cells with a p300-HAT-specific inhibitor C646 (*Ogiwara et al., 2016*) and found that this drug more modestly restored INSM1 and NEU-ROD1, particularly when MSK is co-inhibited (*Figure 4—figure supplement 1b*). Inversely, the expression of NE factors was eliminated in SCLC by inhibition of histone deacetylases (HDACs) using trichostatin A, even in the absence of KRAS$^{G12V}$ induction, suggesting that H3K27ac levels must be restricted to maintain SCLC lineage (*Figure 4d*). Although A-485 treatment did not rescue the ERK-mediated suppression of NE factors in H2107 and H524 cells even when combined with MSK inhibition (*Figure 4—figure supplement 1a*) or with *HES1* knockout (*Figure 4—figure supplement 1c*), these results collectively suggest that constitutively activated ERK suppresses NETFs partly through MSK but mostly via reconfiguration of chromatin structure by CBP/p300 in a subset of SCLC.

## Chromatin accessibility analysis demonstrates enrichment for binding sites of ETS family TFs

The sequencing-based assay for transposase-accessible chromatin (ATAC-seq) (*Buenrostro et al., 2013*) was employed to tease out mechanisms used by ERK and CBP/p300 to reconfigure lung cancer epigenomes. ATAC-seq was performed on H2107, H524, and H82 cells (three biological replicates per condition) with and without treatment with doxycycline to induce KRAS$^{G12V}$ and SCH772984 (ERK inhibitor) for 72 hr, as well as on H82 cells treated with doxycycline in the presence of SB-747651A (MSK/RSK inhibitor), A-485 (CBP/p300 inhibitor), or both. Quality metrics showed good enrichment of accessible chromatin in our ATAC libraries (*Figure 5—figure supplement 1a*) and strong concordance between replicates (Pearson R (*Travis et al., 2015*) >0.90; *Figure 5—figure supplement 1b*). Overall, induction of KRAS$^{G12V}$ expression with doxycycline caused an overall increase in chromatin accessibility (H82: 88 peaks of chromatin accessibility gained; 36 lost; H2107: 38 gained, one lost; H524 638 gained,703 lost). On the contrary, addition of the ERK inhibitor SCH772984 led to a reduction in the number of peaks of accessible chromatin (H82: 131 lost, 58 gained; H2107: 36 lost, one gained; H524: 503 lost, 345 gained; *Figure 5a and b*). The locales of altered accessibility were primarily located in intergenic and intronic regions, in keeping with shifts primarily occurring in regulatory regions, including putative enhancers (*Figure 5—figure supplement 1c and d*). Motif analysis showed that doxycycline treatment led to increased accessibility around ETV1 and ETV4 DNA binding motifs, as well as motifs associated with AP-1 family members, and reduced accessibility at NEUROD1 and ASCL1 motifs (*Figure 5c*). A reversal of this pattern was observed upon treatment with SCH772984 (*Figure 5d*). Motif analyses in individual cell lines showed the same changes in accessibility around these TFs with doxycycline treatment with or without SCH772984 (*Figure 5—figure supplement 2a,b and f–i*). Importantly, the ranked motif order plot with combined inhibition of MSK/RSK and CBP/p300 (*Figure 5—figure supplement 2e*) mimicked that with ERK inhibition (*Figure 5—figure supplement 2b*) in H82 cells. Permutation testing showed

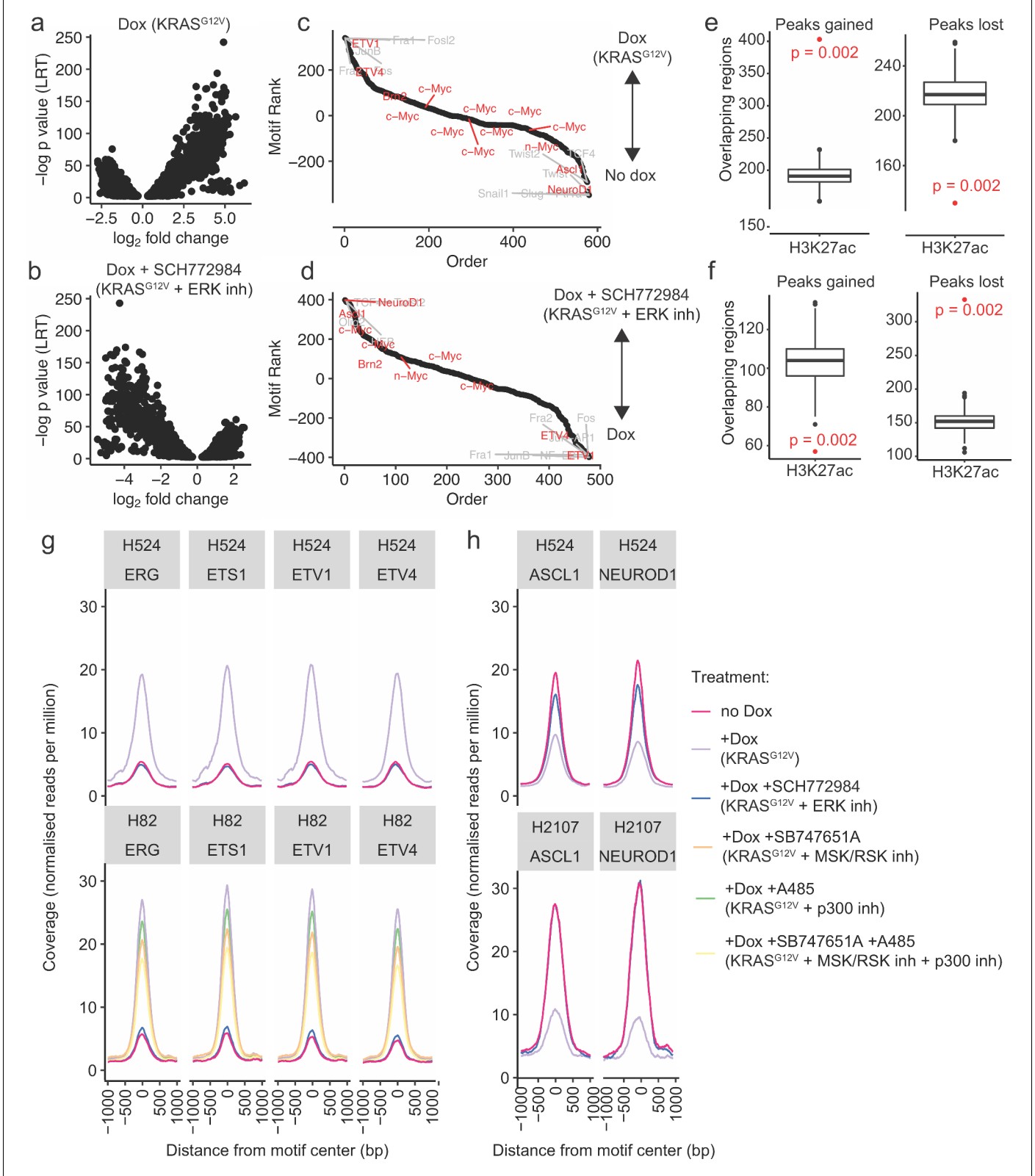

**Figure 5.** ATAC-seq analysis demonstrates chromatin remodeling upon mutant KRAS transduction. (a) Distribution of differentially accessible regions in H524-KRAS^G12V cells upon treatment with doxycycline (dox), and (b) treatment with dox + SCH772984 (an ERK inhibitor [inh]). (c) Ranked list of motif enrichment and depletion over differentially accessible regions in H82-KRAS^G12V, H524-KRAS^G12V, and H2107-KRAS^G12V cells upon dox induction. (d) Ranked list of motif enrichment and depletion over differentially accessible regions in H82-KRAS^G12V, H524-KRAS^G12V, and H2107-KRAS^G12V cells upon

*Figure 5 continued on next page*

Figure 5 continued

dox induction with or without SCH772984 treatment. (e) Permutation testing of co-occupancy of H3K27ac with peaks gained and lost in H524-KRAS^G12V cells upon dox treatment (p value: hypergeometric test). (f) Permutation testing of co-occupancy of H3K27ac with peaks gained and lost in dox-treated H524-KRAS^G12V cells upon SCH772984 treatment (p value: hypergeometric test). (g and h) Occupancy profiles of selected motifs of the (g) ETS family, and (h) selected proneural motifs.

The online version of this article includes the following figure supplement(s) for figure 5:

**Figure supplement 1.** ATAC-seq quality control.
**Figure supplement 2.** Additional ATAC-seq motif analyses.

that peaks gained upon doxycycline induction, with or without SCH779284, were associated with areas of chromatin decorated with H3K27ac (p=0.002, hypergeometric test; *Figure 5e and f*), a histone post-translational modification associated with open chromatin, in control normal human lung. Motif accessibility profiles within differentially accessible regions showed that doxycycline induction led to markedly increased accessibility at the ETV1 and ETV4 binding motifs in H524 and H82 cells (*Figure 5g*). In contrast, chromatin accessibility was reduced at putative binding motifs for neuroendocrine lineage TFs, including ASCL1 and NEUROD1 in H524 and H2107 (*Figure 5h*). No significant changes in overall occupancy were observed at these motifs in H82 cells (*Figure 5—figure supplement 2j*). The overall occupancy profiles of cells treated with SCH772984 most closely resembled those of the untreated cells, in keeping with rescue of the neuroendocrine phenotype.

## ERK activates ETS factors and promotes suppression of NE factors

ATAC-seq demonstrated global chromatin rewiring at ETS transcriptional targets upon KRAS^G12V induction in SCLC cells. Furthermore, ETS TFs – including the PEA3 family of ETS TFs, ETV1, ETV4, and ETV5 – were upregulated at the mRNA level by activation of MAPK (*Supplementary file 1* and *Figure 6a*), suggesting that an ETS TFs-mediated program may play a role in suppressing NE differentiation. Indeed, microarray data of SCLC and LUAD cell lines indicated the clear inverse relationship of expression of PEA3 family ETS TFs with that of NETFs (*Figure 6b*). At the protein level, we found that mutant KRAS upregulates ETV4 in H82 and H524 cells and ETV5 in all the three SCLC lines, which was completely reversed by ERK inhibition with SCH772984 (*Figure 6c*). As ETS TFs bind to a common motif (*Hollenhorst et al., 2011*), we anticipated that overexpression of any one of these proteins in SCLC cells may phenocopy the effects of ERK activation and potentially lead to downregulation of NE factors. To test this, we conditionally expressed ETV1 in the three SCLC cell lines, which led to suppression of specific NETFs – notably ASCL1 in H2107, INSM1 and NEUROD1 in H82, and BRN2 in H524 (*Figure 6d*). Conditional expression of ETV5 also significantly downregulated INSM1 and NEUROD1 in H82, and NEUROD1 in H524 (*Figure 6e*). Furthermore, ETV1- or ETV5-overexpressing cells unexpectedly transformed to an adherent phenotype, with this morphological change most strongly observed in H82 cells, similar to what is observed with KRAS^G12V induction (*Figure 6—figure supplement 1*). Next, we conducted knockdown of *ETV4*, *ETV5*, or both using siRNAs in mutant KRAS-inducible H82 cells and found that *ETV4* knockdown modestly restored suppressed INSM1 while *ETV5* knockdown restored NEUROD1 (*Figure 6f*). Importantly, dual knockdown of *ETV4* and *ETV5* jointly increased INSM1 and CD56 expression, suggesting the functional redundancy of different ETS factors, such that knockdown of a single or two factors is unable to completely mitigate the effects of ERK activation.

CIC/Capicua is a transcriptional repressor of ETV1, ETV4, and ETV5 and a key mediator of MAPK signaling (*Wang et al., 2017*). When the MAPK pathway is activated, ERK and RSK phosphorylate CIC (*Dissanayake et al., 2011*), which is then exported from the nucleus to the cytoplasm and degraded (*Grimm et al., 2012*). We therefore assessed whether CIC is required to maintain the suppression of ETS factors in SCLC and whether its inactivation downstream of ERK modulates NE marker suppression. We confirmed that nuclear CIC was expressed in SCLC cells and downregulated after KRAS^G12V induction, which was restored by ERK inhibition (*Figure 6—figure supplement 2a*). siRNA knockdown of *CIC* led to downregulation of INSM1, NEUROD1, and to a lesser extent BRN2 in H82 cells and potentiated the effects of mutant KRAS induction on NE factor suppression in H82 and H524 cells (*Figure 6—figure supplement 2b*). However, *CIC* knockdown was not sufficient to induce ETV1, ETV4, and ETV5. Likewise, overexpression of CIC did not inhibit suppression of NE

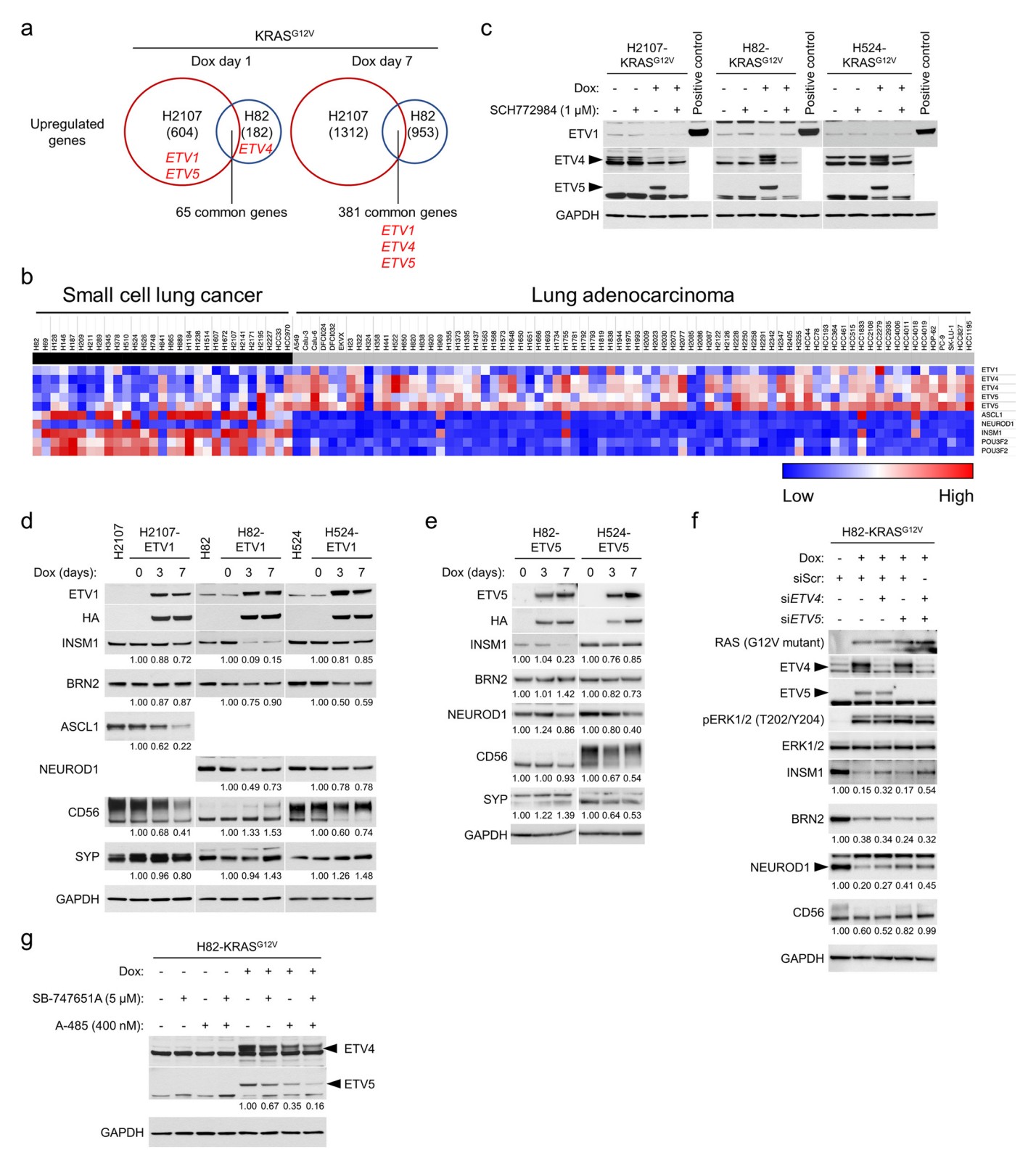

**Figure 6.** The roles of ETS family transcription factors in the regulation of neuroendocrine differentiation in small cell lung cancer cell lines. (a) Upregulated genes by KRAS$^{G12V}$ overexpression for 1 day and 7 days in comparison with a GFP overexpression control in H2107 and H82 cells. The numbers of genes upregulated (>1.5 fold) are indicated. *ETV1*, *ETV4*, and *ETV5* are shown in red. (b) Heat map of the PEA3 family ETS transcription factors (*ETV1*, *ETV4*, and *ETV5*) and neuroendocrine transcription factors (*ASCL1*, *NEUROD1*, *INSM1*, and *POU3F2* [*BRN2*]) in small cell lung cancer and

*Figure 6 continued on next page*

*Figure 6 continued*

lung adenocarcinoma cell lines. Red and blue denote high and low expression, respectively. (c) Western blot showing the effects of ERK inhibition using 1 µM SCH772984 on the expression of ETV1, ETV4, and ETV5 with or without KRAS$^{G12V}$ transduction for 72 hr. Lysates from HA-tagged ETV1-overexpressing H524 cells were used as a positive control for ETV1. GAPDH was used as a loading control. (d) Effects of HA-tagged ETV1 induction as assessed by western blot in H2107, H82, and H524 cells, upon treatment with 100 ng/mL doxycycline (dox) for 3 and 7 days. GAPDH was used as a loading control. Numbers below blots show the amounts of each band relative to the corresponding non-dox-treated control values (set to two in each panel) after normalization to GAPDH. (e) Effects of HA-tagged ETV5 induction as assessed by western blot in H82 and H524 cells, upon treatment with 100 ng/mL dox for 3 and 7 days. GAPDH was used as a loading control. Numbers below blots show the amounts of each band relative to the corresponding non-dox-treated control values (set to one in each cell line panel) after normalization to GAPDH. (f) Western blot showing the effects of KRAS$^{G12V}$ induction and treatment with siRNA pools targeting *ETV4*, *ETV5*, or both on expression of neuroendocrine transcription factors in H82 cells. Cells were treated with 100 ng/mL dox and indicated siRNAs for 72 hr. Scrambled siRNA (siScr) was used as a negative control. GAPDH was used as a loading control. Numbers below blots show the amounts of each band relative to the corresponding non-dox-treated and siScr-treated control values (set to 1) after normalization to GAPDH. (g) Western blot showing the effects of MSK/RSK and/or CBP/p300 inhibition on KRAS$^{G12V}$-mediated expression of ETV4 and ETV5 in H82 cells. Cells were treated with 5 µM SB-747651A (a MSK-RSK inhibitor) and/or 400 nM A-485 (a CBP/p300 inhibitor) as well as 100 ng/mL dox for 72 hr. GAPDH was used as a loading control. Numbers below the ETV5 blots indicate the amounts of ETV5 relative to the dox-treated and non-drug-treated (DMSO-treated) control values (set to lane 5) after normalization to GAPDH. Immunoblots are representative of at least two biological replicates.

The online version of this article includes the following figure supplement(s) for figure 6:

**Figure supplement 1.** ETV1-induced phenotypic change in the growing pattern in small cell lung cancer cell lines.

**Figure supplement 2.** The roles of CIC in the regulation of neuroendocrine differentiation in small cell lung cancer cell lines.

**Figure supplement 3.** Western blot showing the effects of different combinations of inhibitors targeting MSK/RSK, CBP/p300, or ERG on expression of neuroendocrine transcription factors that are repressed by KRAS$^{G12V}$ transduction in small cell lung cancer cells.

**Figure supplement 4.** Effects of *RB1* knockout on sensitivity and resistance mechanisms to osimertinib in *TP53* and *EGFR* double-mutant lung adenocarcinoma cell lines.

**Figure supplement 5.** Cell morphology and profiling of neuroendocrine transcription factors in H1975 cells cultured in stem cell culture media (SCCM).

factors after KRAS$^{G12V}$ induction (*Figure 6—figure supplement 2c*), as putatively inactivated following immediate phosphorylation by activated ERK. These results suggest that ERK-mediated upregulation of the PEA3 family of ETS TFs does not occur via CIC inhibition in SCLC cells. However, as we demonstrated that the PEA3 family of ETS TFs are – at least in part – a mediator of ERK-induced suppression of NETFs, we next asked if CBP/p300 activation by ERK regulates PEA3 TFs in a CIC-independent manner. Indeed, we found that A-485 treatment downregulates ERK-induced ETV4 and ETV5 in H82-KRAS cells but not in H2107- and H524-KRAS cells, providing a potential biological explanation why CBP/p300 inhibition rescues ERK-mediated suppression of NE differentiation only in H82 cells (*Figure 6g* and *Figure 4—figure supplement 1a*).

Lastly, it has been reported that oncogenic fusion proteins produced by chromosomal translocations are the major mechanism of genetic activation of ETS family proteins in cancer. In prostate cancer, the ETS family members *ERG* as well as *ETV1* are commonly rearranged (*Tomlins et al., 2005*) and ectopic ERG expression by *TMPRSS2-ERG* fusion blocks NE differentiation (*Mounir et al., 2015*). Based on these findings, we treated KRAS$^{G12V}$-inducible SCLC cells with an ERG inhibitor, ERGi-USU (*Mohamed et al., 2018*), with or without MSK/RSK inhibition (*Figure 6—figure supplement 3a*). ERG was not basally expressed in the three SCLC cell lines but was induced by KRAS$^{G12V}$ in H82 and H524 cells. Inhibition of ERG in H82 cells after KRAS$^{G12V}$ activation provided modest restoration of INSM1, BRN2, and NEUROD1 at the optimal concentration of 0.6 µM, which synergized with MSK/RSK co-inhibition. Interestingly, treatment of H82-KRAS$^{G12V}$ cells with different combinations of inhibitors targeting MSK/RSK, CBP/p300, or ERG revealed that suppression of MYC by KRAS$^{G12V}$ was well rescued by combined MSK/RSK and ERG inhibition that did not rescue suppressed expression of key NETFs, suggesting that oncogene-mediated ERK activation in SCLC modulates essential TFs through multiple regulatory mechanisms (*Figure 6—figure supplement 3b*). It should be noted that ERGi-USU treatment also inhibited ETV5 expression in a dose-dependent manner in H2107 and H82 cells (*Figure 6—figure supplement 3a*), showing that this compound may work broadly on ETS factors and not exclusively through ERG. Together, these results suggest that ERK-induced ETS factor expression suppresses NE lineage factors in SCLC and that induction of the PEA3 family of ETS TFs is mediated by the HAT activity of CBP/p300 in H82 cells but not in H2107 and H524 cells.

## CIC inactivation in *EGFR*-mutant LUAD upon osimertinib resistance suppresses SCLC transformation in p53/RB-inactivated cells

Using the information obtained from expression of mutant EGFR and KRAS in SCLC, we aimed to assess the potential clinical importance of these mechanisms in driving the transformation of LUAD to SCLC during EGFR-TKI resistance. Dual p53/RB inactivation is ubiquitous in SCLC (*Peifer et al., 2012*; *George et al., 2015*), and *EGFR*-mutant LUADs with p53/RB loss are more likely to undergo SCLC transformation after TKI treatment (*Offin et al., 2019*; *Niederst et al., 2015*; *Lee et al., 2017*). Furthermore, p53/RB inactivation in androgen receptor (AR)-dependent prostate luminal epithelial tumors increases SOX2 expression and causes lineage shift into basal-like or NE tumors that are AR-independent (*Mu et al., 2017*). Therefore, we tested whether this scenario is also applicable in EGFR-dependent LUAD cells. We selected two *TP53/EGFR* double-mutant and *RB1* wild-type cell lines, PC9 and H1975, and performed *RB1* knockout through CRISPR/Cas9, establishing *TP53/RB1/EGFR* triple-mutant clones (*Figure 6—figure supplement 4a*). We then treated these clones, along with *RB1*-proficient control cells, with osimertinib to assess the influence of EGFR/MAPK inactivation on NE differentiation in the p53/RB-deficient background. Unlike the prostate cancer scenario, deregulation of SOX2 was not observed following osimertinib treatment, irrespective of the *RB1* status (*Figure 6—figure supplement 4b*). In addition, NE factors were not induced in the triple-mutant clones, suggesting that the LUAD lineage is more strictly maintained than the lineage of AR-dependent prostate cancer in the context of dual p53/RB inactivation, confirming a previous study (*Niederst et al., 2015*).

We then attempted to force SCLC transformation from these triple-mutant LUAD cells by long-term exposure to osimertinib. Although *RB1* knockout shifted the initial IC50 values to the drug with statistical significance in H1975 cells (*Figure 6—figure supplement 4c*), the effects were modest. We derived resistant cells through two methods – dose escalation or with an initial high-dose – and confirmed insensitivity to osimertinib in comparison to equally passaged control cells (*Figure 6—figure supplement 4d*). As resistant cells remained adherent, we asked if EGFR-independent reactivation of ERK inhibited NE trans-differentiation in these cells and assessed acquired genetic alterations using MSK-IMPACT targeted genomic profiling (*Figure 6—figure supplement 4e*). This revealed mutations and amplifications that reactivate MAPK pathway in resistant clones compared to their parental counterparts, including *ARAF*, *NRAS*, and *ERBB4* mutations as well as amplifications of *MAPK3* and *NRAS* in three of 12 resistant clones. Correspondingly, pERK was still detectable in the majority of resistant clones in the presence of osimertinib (*Figure 6—figure supplement 4f*), unlike parental cell lines with acute treatment (*Figure 6—figure supplement 4b*). Western blot analysis also confirmed no induced expression of the main NE factors in resistant cells (*Figure 6—figure supplement 4f*). Importantly, *CIC* mutations that bypass the requirement for upstream MAPK pathway reactivation were recurrently identified in H1975 resistant clones (*Figure 6—figure supplement 4e*), which was validated by western blot (*Figure 6—figure supplement 4g*). Among the PEA3 family of ETS TFs, ETV5 was most prominently upregulated in osimertinib-resistant clones harboring acquired *CIC* alterations (*Figure 6—figure supplement 4g*). These data collectively suggest that recurrently observed resistance mechanisms that reactivate the ERK/CIC/ETS axis might suppress the NE differentiation program during chronic inhibition of drivers even in the context of p53/RB loss.

## Inhibition of ERK and MSK/RSK in stem cell culture media induces neuronal-like differentiation and suppression of EGFR in *EGFR*-mutant lung adenocarcinoma

Based on our findings that ERK, MSK/RSK, and CBP/p300 play critical roles in the regulation of NETFs in SCLC cell lines, we treated *EGFR/TP53/RB1* triple-mutant H1975 cells with inhibitors for EGFR, ERK, MSK/RSK, and/or CBP/p300 to inhibit effectors that suppress NE differentiation with the anticipation that it would eventually cause histological transformation into SCLC. To this end, we cultured cells using stem cell culture media (SCCM) as well as RPMI 1640, as a previous study used SCCM in conjunction with genetic manipulations to reprogram normal human lung epithelial cells to neuroendocrine lineage (*Park et al., 2018*). When cultured in SCCM, H1975 cells grew in suspension as floating clusters (*Figure 6—figure supplement 5a*). Interestingly, inhibition of ERK and MSK/RSK in SCCM inhibited the phenotypic change into floating suspension. Furthermore, cells developed a neuronal-like appearance showing bipolar or multipolar cells with axonal processes after combined

inhibition of ERK and MSK/RSK regardless of *RB1* status (*Figure 6—figure supplement 5a*), which was coupled with suppression of EGFR (*Figure 6—figure supplement 5b*). Immunoblotting showed that phospho-AKT was highly upregulated after combined inhibition of ERK and MSK/RSK (*Figure 6—figure supplement 5b*), highlighting the potential importance of AKT signaling in cell morphology and growing phenotype. However, the triple-mutant cells, including the neuronal-like cells, showed no induction of NETFs over 3 (*Figure 6—figure supplement 5b*) or 7 (*Figure 6—figure supplement 5c*) days of culture with different combinations of inhibitors, both in normal media and SCCM.

## Discussion

In addition to differences in RB and p53 status, SCLC differs from LUAD in the absence of activating mutations in the MAPK pathway. Furthermore, LUAD tumors that transform to SCLC during TKI resistance lack MAPK signaling, suggesting an inverse relationship between the activity of this pathway and NE differentiation. Here, we have investigated how constitutively activated MAPK signaling driven by exogenous expression of mutant KRAS or EGFR affects the NE program in SCLC cell lines. We found that activation of the downstream signaling node of MAPK pathway, ERK2, suppresses the expression of crucial NE lineage master regulators in SCLC. Furthermore, we found that chromatin regions bound by NETFs were rendered less accessible by activated ERK and that this chromatin remodeling was associated with activation of CBP/p300 HAT activity and corresponding changes in H3K27ac-marked regions. ETS transcription factors were also upregulated as a result of this remodeling, which we showed was an additional mechanism of ERK-mediated suppression of NETFs. Together, this suggests that an ERK-MSK/RSK-CBP/p300-ETS axis, acting through multiple points, synergizes to suppress NE differentiation, providing biological rationale for the absence of MAPK pathway activation in SCLC (*Figure 7a*).

However, it should be noted that this mechanism was not applicable across all cell lines we assessed, indicative of the multifactorial mechanisms by which ERK suppresses NE differentiation in a context-dependent manner. This likely relates to differences in expression of master NE regulators such as ASCL1 and NEUROD1, mutation status of epigenomic modifiers including *CREBBP* (*Jia et al., 2018*) and *EP300*, basal ERK activity, cell-specific mechanisms in MAPK pathway feedback loops, and cross-talk with other signaling pathways. Indeed, H2107 and H524 harbor mutations in *EP300* while H82 is *EP300* wild-type (*Figure 7b*), which might contribute to relatively low NE differentiation in H82 (*Figure 1g*) and its increased permissiveness to ERK-induced CBP/p300 HAT activity and NE factor suppression. Furthermore, MYC status has been attributed to phenotypic diversity of SCLCs, with MYCL- and MYC-driven SCLC cell lines differing in numerous aspects from transcriptional signatures, super enhancer usage, metabolic regulation, and therapeutic response (*Ireland et al., 2020*). As H2107 expresses MYCL while H82 and H524 are MYC high (*Figure 7b*), this could also underlie the differential response to ERK induction, such as those seen on cell growth and the involvement of CBP/p300 activation and H3K27ac in mediating NETF suppression. Nevertheless, our work clearly demonstrated that ERK2 kinase activity plays a central role in inhibition of NE differentiation in all SCLC cell lines assessed. As ERK2 is known to directly shift transcriptional machinery in a kinase-dependent manner (*Hamilton et al., 2019*; *Göke et al., 2013*), direct enhancer regulation by ERK2 may be involved in our model, with ERK-mediated chromatin remodeling independent of CBP/p300 activity also playing an important role in specific contexts (*Figure 7c*).

Our findings provide biological bases for the mutual exclusivity between gene alterations in MAPK pathway and NE differentiation in lung cancer. We showed that mutant-KRAS induction in SCLC more robustly suppresses NE differentiation and affects growth morphology than mutant-EGFR induction. The former is explained by the different potency of ERK activation between these two LUAD oncoproteins and is in line with the fact that *KRAS* mutations are not detected in SCLC specimens (*Peifer et al., 2012*; *George et al., 2015*). Moreover, a recent study highlighted the potentially deleterious effect of activated ERK in NE tumors by demonstrating that loss of *ERK2* and negative ERK2 expression are specific features of the neuroendocrine carcinoma component of gastric mixed adenoneuroendocrine carcinoma (*Sun et al., 2020*). Although one major exception for this scenario is *FGFR* amplification that is observed in 6% of SCLC cases (*Peifer et al., 2012*; *George et al., 2015*) and is anticipated to activate the MAPK pathway, another recent study showed

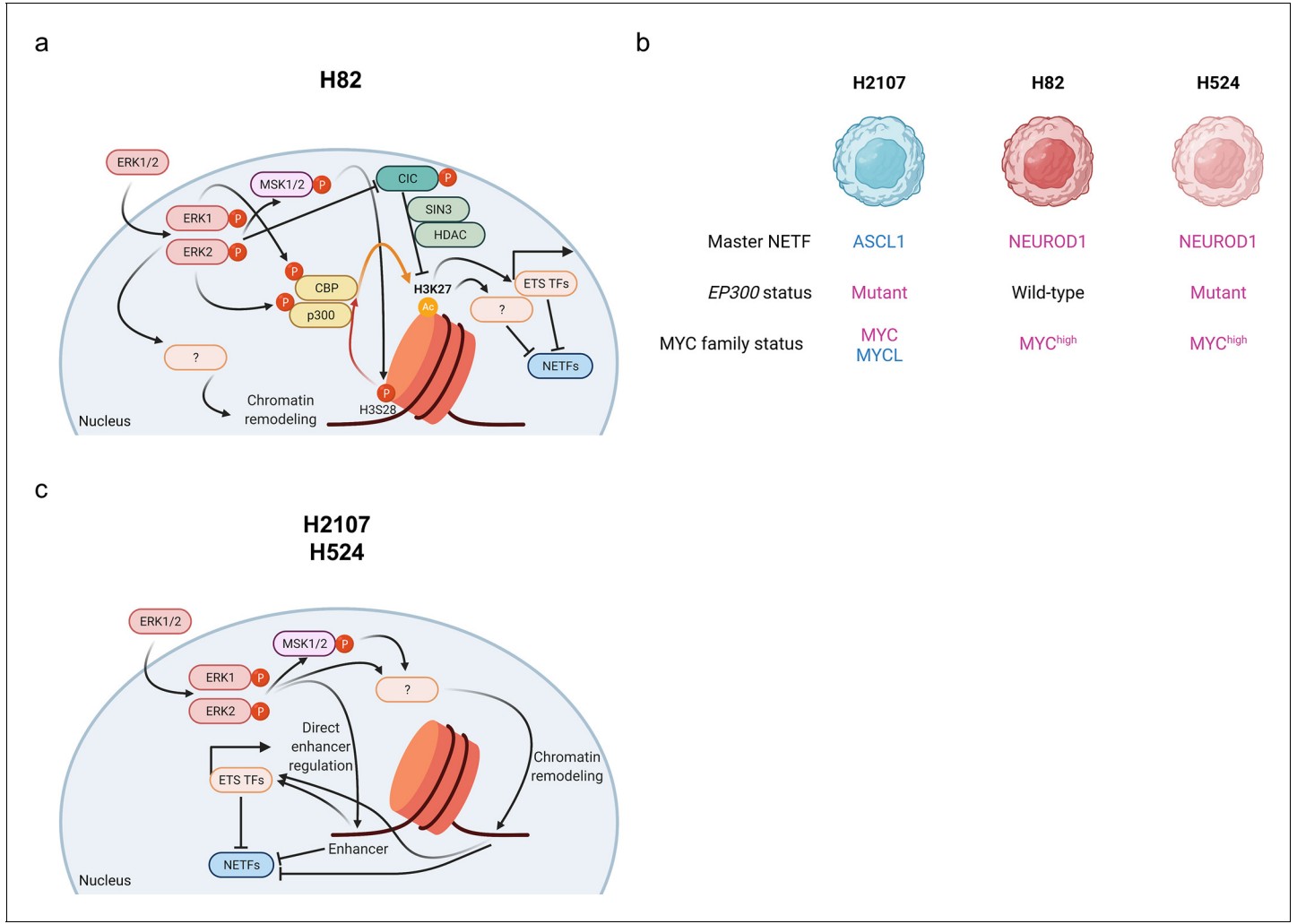

**Figure 7.** Schematic representation of the mechanism of ERK-mediated repression of neuroendocrine differentiation through chromatin dysregulation in small cell lung cancer. (a) and (c) Different mechanisms according to cell lines ((a), for H82; (c), for H2107 and H524) are depicted. (b) Overview of characteristics of cell lines used in this study. Illustration was created with BioRendrer.com. HDAC, histone deacetylase; TFs, transcription factors; NETFs, neuroendocrine transcription factors.

that FGFR1 is disadvantageous for the development of typical central SCLC with NE differentiation using a $Rb1^{flox/flox}$;$Trp53^{flox/flox}$;$LSL\text{-}Fgfr1^{K656E}$ mouse model (*Ferone et al., 2020*).

The recurrently observed inactivating gene alterations of *CREBBP/EP300* (*Peifer et al., 2012*; *George et al., 2015*; *Rudin et al., 2012*) in SCLC, as well as in *EGFR*-mutant LUAD tumors that subsequently undergo TKI-induced SCLC transformation (*Offin et al., 2019*), suggest that a global or local increase in H3K27ac is incompatible with SCLC. Indeed, HDAC inhibitor treatment, which was shown to inhibit SCLC growth (*Jia et al., 2018*), strongly suppressed NETFs in SCLC cells (*Figure 4d*), despite potential specificity issues related to these inhibitors. However, we were unable to conclusively demonstrate that CBP/p300 inhibition promotes SCLC transformation in conjunction with loss of p53 and RB in EGFR-TKI-treated *EGFR*-mutant LUAD cells as hypothesized. Given that CBP/p300 involvement was observed in only H82 cells, other mechanisms may be required to cause SCLC transformation of *EGFR*-mutant LUAD. It may also be attributed to the concentrations of A-485 (100–400 nM) and C646 (10 μM) tested. Although these ranges of the drugs most potently restored NETFs which were suppressed in mutant KRAS-transduced H82 cells, complete elimination of H3K27ac by higher-doses of A-485 paradoxically downregulated NETFs (*Figure 4—figure supplement 1a*), highlighting the challenges of using A-485 to induce NETFs, particularly when combined with additional compounds that inhibit RTK-MAPK components that also downregulate H3K27ac

levels. The optimal decrease of H3K27ac level required to restore suppressed NETFs by ERK activation is likely cell-line-specific and context-dependent, making it difficult to employ genomic methodologies such as CRIPSR and shRNA for such rescue experiments. Nevertheless, our results imply that the use of HDAC inhibitors during treatment with EGFR-TKIs might be an option to prevent SCLC transformation in *EGFR*-mutant LUAD at a high risk of SCLC transformation.

Our results revealed that ERK-mediated upregulation of the PEA3 family of ETS TFs as well as ERG suppresses NE differentiation in SCLC. Because ETS TFs recognize the same ETS DNA-binding motif (*Hollenhorst et al., 2011*), upregulation of these factors is likely to work in collaboration. However, it is notable that even ETV1 or ETV5 overexpression alone or knockdown of *ETV4* or *ETV5* alone showed effects on NETFs expression. Surprisingly, a well-described upstream repressor of these factors, CIC, was not involved in the ERK-mediated upregulation of ETV4 and ETV5 in SCLC cells. Similar to the observation of NOTCH-independent HES1 upregulation observed in the present and a previous (*Stockhausen et al., 2005*) studies, oncogene-activated ERK in SCLC induces ETS TFs independent of CIC inhibition, which will require further investigation. Nonetheless, we found that *CIC* knockdown suppressed NETFs in H82 cells. Given that CBP/p300 inhibition restored ERK-mediated suppression of NETFs only in H82 cells, an ETS-independent repressor function of CIC, perhaps the previously reported role of interacting with the SIN3 deacetylation complex and recruiting HDACs (*Weissmann et al., 2018*), may be involved in this process (*Figure 7a*).

In summary, we provide the first reported biological rationale for why alterations in MAPK pathway are rarely found in SCLC and describe the molecular underpinnings of how the central node in this pathway, ERK2, suppresses the NE differentiation program. However, we could not force *EGFR*-mutated LUAD to transform to SCLC as a resistance mechanism to osimertinib, despite our attempts to reverse engineer this process based on this knowledge. Our strategy using extensively passaged established cell lines might not be suitable to cause lineage transformation, and applications of patient-derived tumor organoids or genetically engineered mouse models that potentially possess the ability of differentiation into different lineage might be promising options in the future. In addition, there are a number of reported mechanisms of SCLC transformation in prostate cancer (*Mu et al., 2017*; *Bishop et al., 2017*; *Guo et al., 2019*; *Cerasuolo et al., 2015*), suggesting genetic heterogeneity may complicate efforts to model this process in lung cancer. We propose that SCLC transformation from LUAD requires specific conditions in the background of p53/RB loss, including lack of both second site *EGFR* mutations and bypass pathway mutations that reactivate MAPK signaling and ETS TFs, and alterations in epigenetic modifiers such as CBP/p300 to reduce H3K27ac levels and alter chromatin accessibility, rendering gene expression suitable for NE differentiation. While this work provides fundamental information regarding lineage plasticity in lung cancer, further studies are required to identify novel therapeutic approaches targeting histological trans-differentiation between LUAD and SCLC in the context of EGFR-TKI resistance.

## Materials and methods

### Key resources table

| Reagent type (species) or resource | Designation | Source or reference | Identifiers | Additional information |
|---|---|---|---|---|
| Cell line (*Homo sapiens*) | NCI-H2107 | ATCC | CRL-5983_FL | |
| Cell line (*Homo sapiens*) | NCI-H82 | ATCC | HTB-175 | |
| Cell line (*Homo sapiens*) | NCI-H524 | ATCC | CRL-5831 | |
| Cell line (*Homo sapiens*) | NCI-H526 | ATCC | CRL-5811 | |
| Cell line (*Homo sapiens*) | PC-9 | Adi Gazdar | | |
| Cell line (*Homo-sapiens*) | NCI-H1650 | ATCC | CRL-5883 | |

*Continued on next page*

*Continued*

| Reagent type (species) or resource | Designation | Source or reference | Identifiers | Additional information |
|---|---|---|---|---|
| Cell line (*Homo sapiens*) | NCI-H1975 | ATCC | CRL-5908 | |
| Cell line (*Homo-sapiens*) | HCC827 | ATCC | CRL-2868 | |
| Cell line (*Homo sapiens*) | HCC2279 | ATCC | CRL-2870 | |
| Cell line (*Homo sapiens*) | HCC2935 | ATCC | CRL-2869 | |
| Cell line (*Homo sapiens*) | HCC4006 | ATCC | CRL-2871 | |
| Cell line (*Homo sapiens*) | HCC4011 | Adi Gazdar | RRID:CVCL_S700 | |
| Cell line (*Homo-sapiens*) | NCI-H23 | ATCC | CRL-5800 | |
| Cell line (*Homo sapiens*) | NCI-H1792 | ATCC | CRL-5895 | |
| Cell line (*Homo sapiens*) | A549 | ATCC | CCL-185 | |
| Cell line (*Homo sapiens*) | NCI-H358 | ATCC | CRL-5807 | |
| Cell line (*Homo sapiens*) | NCI-H1395 | ATCC | CRL-5868 | |
| Cell line (*Homo sapiens*) | NCI-H2347 | ATCC | CRL-5942 | |
| cell line (*Homo sapiens*) | NCI-H460 | ATCC | HTB-177 | |
| Cell line (*Homo sapiens*) | NCI-H1155 | ATCC | CRL-5818 | |
| Transfected construct (human) | lentiCRISPRv2 | Addgene | RRID:Addgene_52961 | Lentiviral construct to transfect and express hSpCas9 and sgRNA. |
| Transfected construct (human) | pcDNA4/TO | *Wong et al., 2019* PMID:30093628 | | Tetracycline-regulated mammalian expression vector |
| Transfected construct (human) | siRNA to *MAPK3* (SMARTpool) | Horizon Discovery | L-003592–00 | transfected construct (human) |
| Transfected construct (human) | siRNA to *MAPK1* (SMARTpool) | Horizon Discovery | L-003555–00 | transfected construct (human) |
| Transfected construct (human) | siRNA to *REST* (SMARTpool) | Horizon Discovery | L-006466–00 | transfected construct (human) |
| Transfected construct (human) | siRNA to *CIC* (SMARTpool) | Horizon Discovery | L-015185–01 | transfected construct (human) |
| Transfected construct (human) | siRNA to *ETV4* (SMARTpool) | Horizon Discovery | L-004207–00 | transfected construct (human) |
| Transfected construct (human) | siRNA to *ETV5* (SMARTpool) | Horizon Discovery | L-008894–00 | transfected construct (human) |
| Antibody | anti-phospho-EGFR (Tyr1068) (Rabbit polyclonal) | Cell Signaling Technology | Cat. #: 2234 | WB (1:1000) |
| Antibody | anti-EGFR (Rabbit polyclonal) | Cell Signaling Technology | Cat. #: 2232 | WB (1:1000) |
| Antibody | anti-Brn2/POU3F2 (D2C1L) (Rabbit monoclonal) | Cell Signaling Technology | Cat. #: 12137 | WB (1:1000) |

*Continued on next page*

*Continued*

| Reagent type (species) or resource | Designation | Source or reference | Identifiers | Additional information |
|---|---|---|---|---|
| Antibody | anti-INSM1 (A-8) (Mouse monoclonal) | Santa Cruz Biotechnology | Cat. #: sc-271408 | WB (1:1000) |
| Antibody | anti-MASH1 (24B72D11.1) (Mouse monoclonal) | BD Pharmingen Inc | Cat. #: 556604 | WB (1:1000) |
| Antibody | anti-NeuroD1 (D35G2) (Rabbit monoclonal) | Cell Signaling Technology | Cat. #: 4373 | WB (1:1000) |
| Antibody | anti-NCAM1 (CD56) (123C3) (Mouse monoclonal) | Cell Signaling Technology | Cat. #: 3576 | WB (1:1000) |
| Antibody | anti-Synaptophysin (Rabbit polyclonal) | Cell Signaling Technology | Cat. #: 4329 | WB (1:1000) |
| Antibody | anti-phospho-p44/42 MAPK (Thr202/Tyr204) (Rabbit polyclonal) | Cell Signaling Technology | Cat. #: 9101 | WB (1:1000) |
| Antibody | anti-p44/42 MAPK (137F5) (Rabbit monoclonal) | Cell Signaling Technology | Cat. #: 4695 | WB (1:1000) |
| Antibody | anti-Akt (Ser473) (D9E) (Rabbit monoclonal) | Cell Signaling Technology | Cat. #: 4060 | WB (1:2000) |
| Antibody | anti-Akt (pan) (C67E7) (Rabbit monoclonal) | Cell Signaling Technology | Cat. #: 4691 | WB (1:1000) |
| Antibody | anti-Ras (G12V mutant specific) (D2H12) (Rabbit monoclonal) | Cell Signaling Technology | Cat. #: 14412 | WB (1:1000) |
| Antibody | anti-Ras (D2C1) (Rabbit monoclonal) | Cell Signaling Technology | Cat. #: 8955 | WB (1:1000) |
| Antibody | anti-c-Myc (D84C12) (Rabbit monoclonal) | Cell Signaling Technology | Cat. #: 5605 | WB (1:1000) |
| Antibody | anti-cleaved PARP (Asp214) (D64E10) (Rabbit monoclonal) | Cell Signaling Technology | Cat. #: 5625 | WB (1:1000) |
| Antibody | anti-YAP (D8H1X) (Rabbit monoclonal) | Cell Signaling Technology | Cat. #: 14074 | WB (1:1000) |
| Antibody | anti-phospho-p90RSK (Ser380) (D3H11) (Rabbit monoclonal) | Cell Signaling Technology | Cat. #: 11989 | WB (1:1000) |
| Antibody | anti-RSK1/RSK2/RSK3 (D7A2H) (Rabbit monoclonal) | Cell Signaling Technology | Cat. #: 14813 | WB (1:1000) |
| Antibody | anti-phospho-CREB (Ser133) (1B6) (Mouse monoclonal) | Cell Signaling Technology | Cat. #: 9196 | WB (1:1000) |
| Antibody | anti-CREB (48H2) (Rabbit monoclonal) | Cell Signaling Technology | Cat. #: 9197 | WB (1:1000) |
| Antibody | anti-Notch1 (D1E11) (Rabbit monoclonal) | Cell Signaling Technology | Cat. #: 3608 | WB (1:1000) |
| Antibody | anti-cleaved Notch1 (Val1744) (D3B8) (Rabbit monoclonal) | Cell Signaling Technology | Cat. #: 4147 | WB (1:1000) |
| Antibody | anti-Notch2 (D76A6) (Rabbit monoclonal) | Cell Signaling Technology | Cat. #: 5732 | WB (1:1000) |

*Continued on next page*

*Continued*

| Reagent type (species) or resource | Designation | Source or reference | Identifiers | Additional information |
|---|---|---|---|---|
| Antibody | anti-HES1 (D6P2U) (Rabbit monoclonal) | Cell Signaling Technology | Cat. #: 11988 | WB (1:1000) |
| Antibody | anti-Sox2 (D6D9) (Rabbit monoclonal) | Cell Signaling Technology | Cat. #: 3579 | WB (1:1000) |
| Antibody | anti-Sox9 (D8G8H) (Rabbit monoclonal) | Cell Signaling Technology | Cat. #: 82630 | WB (1:1000) |
| Antibody | anti-H3K4me3 (C42D8) (Rabbit monoclonal) | Cell Signaling Technology | Cat. #: 9751 | WB (1:1000) |
| Antibody | anti-H3K9me3 (D4W1U) (Rabbit monoclonal) | Cell Signaling Technology | Cat. #: 13969 | WB (1:1000) |
| Antibody | anti-H3K9ac (C5B11) (Rabbit monoclonal) | Cell Signaling Technology | Cat. #: 9649 | WB (1:1000) |
| Antibody | anti-H3K14ac (D4B9) (Rabbit monoclonal) | Cell Signaling Technology | Cat. #: 7627 | WB (1:1000) |
| Antibody | anti-H3K27ac (D5E4) (Rabbit monoclonal) | Cell Signaling Technology | Cat. #: 8173 | WB (1:1000) |
| Antibody | anti-H3S10ph (Ser10) (Rabbit polyclonal) | Cell Signaling Technology | Cat. #: 9701 | WB (1:1000) |
| Antibody | anti-H3S28ph (Ser28) (Rabbit polyclonal) | Cell Signaling Technology | Cat. #: 9713 | WB (1:1000) |
| Antibody | anti-Histone H3 (D1H2) (Rabbit monoclonal) | Cell Signaling Technology | Cat. #: 4499 | WB (1:2000) |
| Antibody | anti-CIC (Rabbit polyclonal) | Thermo Fisher Scientific | Cat. #: PA1-46018 | WB (1:1000) |
| Antibody | anti-HA-Tag (C29F4) (Rabbit monoclonal) | Cell Signaling Technology | Cat. #: 3724 | WB (1:1000) |
| Antibody | anti-ETV1 (Rabbit polyclonal) | Thermo Fisher Scientific | Cat. #: PA5-41484 | WB (1:1000) |
| Antibody | anti-Pea3 (Rabbit polyclonal) | Abcam | Cat. #: ab189826 | WB (1:1000) |
| Antibody | anti-ERM/Etv5 (Rabbit polyclonal) | Abcam | Cat. #: ab102010 | WB (1:1000) |
| Antibody | anti-ERG (A7L1G) (Rabbit monoclonal) | Cell Signaling Technology | Cat. #: 97249 | WB (1:1000) |
| Antibody | anti-Rb (4H1) (Mouse monoclonal) | Cell Signaling Technology | Cat. #: 9309 | WB (1:2000) |
| Antibody | anti-β-Actin (D6A8) (Rabbit monoclonal) | Cell Signaling Technology | Cat. #: 12620 | WB (1:1000) |
| Antibody | anti-GAPDH (0411) (Mouse monoclonal) | Santa Cruz Biotechnology | Cat. #: sc-47724 | WB (1:3000) |
| Recombinant DNA reagent | pInducer20 (plasmid) | Addgene | RRID:Addgene_44012 | Tet-inducible lentiviral vector for ORF expression |
| Recombinant DNA reagent | pInducer20-GFP (plasmid) | *Unni et al., 2015* PMID:26047463 | | GFP version of pInducer20 |
| Recombinant DNA reagent | pInducer20-EGFR$^{L858R}$ (plasmid) | *Unni et al., 2015* PMID:26047463 | | EGFR$^{L858R}$ version of pInducer20 |
| Recombinant DNA reagent | pInducer20-KRAS$^{G12V}$ (plasmid) | *Unni et al., 2015* PMID:26047463 | | KRAS$^{G12V}$ version of pInducer20 |
| Recombinant DNA reagent | pInducer20-ETV1 (plasmid) | This paper – Materials and methods Section | Lockwood Lab | ETV1 version of pInducer20 |
| Recombinant DNA reagent | pInducer20-ETV5 (plasmid) | This paper – Materials and methods Section | Lockwood Lab | ETV5 version of pInducer20 |

*Continued on next page*

*Continued*

| Reagent type (species) or resource | Designation | Source or reference | Identifiers | Additional information |
|---|---|---|---|---|
| Recombinant DNA reagent | pcDNA4/TO/FLAG-CIC-L | *Wong et al., 2019* PMID:30093628 | | Tetracycline-regulated CIC-L expression vector |
| Recombinant DNA reagent | pcDNA4/TO/FLAG-CIC-S | *Wong et al., 2019* PMID:30093628 | | Tetracycline-regulated CIC-S expression vector |
| Recombinant DNA reagent | pcDNA4/TO/FLAG-CIC-S$^{V41G}$ | *Wong et al., 2019* PMID:30093628 | | Tetracycline-regulated CIC-S$^{V41G}$ expression vector |
| Sequence-based reagent | sg*HES1*-1 | This paper | sgRNA sequence | GTGCTGGGGAAGTACCGAGC |
| Sequence-based reagent | sg*HES1*-2 | This paper | sgRNA sequence | GGTATTAACGCCCTCGCACG |
| Sequence-based reagent | sg*SOX9*-1 | This paper | sgRNA sequence | CAAAGGCTACGACTGGACGC |
| Sequence-based reagent | sg*SOX9*-2 | This paper | sgRNA sequence | AGGTGCTCAAAGGCTACGAC |
| Sequence-based reagent | sg*RB1*-1 | PMID:26314710 | sgRNA sequence *Nicolay et al., 2015* | GCTCTGGGTCCTCCTCAGGA |
| Sequence-based reagent | siRNA: non-targeting control | Horizon Discovery | D-001810–10 | transfected construct (human) |
| Commercial assay or kit | Proteome Profiler Human Phospho-Kinase Array Kit | R and D Systems | Cat. #: ARY003B | |
| Commercial assay or kit | *Quick*-RNA Miniprep Kit | Zymo Research | Cat. #: R1054 | |
| Commercial assay or kit | DNeasy Blood and Tissue Kit | Qiagen | Cat. #: 69506 | |
| Commercial assay or kit | High-Capacity RNA-to-cDNA Kit | Thermo Fisher Scientific | Cat. #: 4387406 | |
| Commercial assay or kit | Gateway LR Clonase II enzyme mix | Thermo Fisher Scientific | Cat. #: 11791020 | |
| Commercial assay or kit | TaqMan Gene Expression Assay Mix for *REST* | Thermo Fisher Scientific | Cat. #: Hs05028212_s1 | |
| Commercial assay or kit | TaqMan Gene Expression Assay Mix for ACTB | Thermo Fisher Scientific | Cat. #: Hs99999903_m1 | |
| Commercial assay or kit | NE-PER Nuclear and Cytoplasmic Extraction Reagents | Thermo Fisher Scientific | Cat. #: 78833 | |
| Chemical compound, drug | Doxycycline hyclate | Sigma Aldrich | D9891 | |
| Chemical compound, drug | SCH772984 | Selleck Chemicals | S7101 | |
| Chemical compound, drug | MK-2206 2HCl | Selleck Chemicals | S1078 | |
| Chemical compound, drug | RO4929097 | Selleck Chemicals | S1575 | |
| Chemical compound, drug | SB-747651A dihydrochloride | Tocris Bioscience | 4630 | |
| Chemical compound, drug | A-485 | Tocris Bioscience | 6387 | |
| Chemical compound, drug | Trichostatin A | Selleck Chemicals | S1045 | |
| Chemical compound, drug | C646 | Selleck Chemicals | S7152 | |

*Continued*

| Reagent type (species) or resource | Designation | Source or reference | Identifiers | Additional information |
|---|---|---|---|---|
| Chemical compound, drug | ERGi-USU | Tocris Bioscience | 6632 | |
| Chemical compound, drug | Osimertinib | Selleck Chemicals | S7297 | |
| Software, algorithm | R software | R Foundation for Statistical Computing | Version 3.6.1 | |
| Software, algorithm | GSEA software | PMID:16199517 | Version 4.0.3 | |
| Software, algorithm | GraphPad Prism | GraphPad Software | Version 8.2.1 | |

## Cell lines and cell culture

Cells were cultured at 37°C with 5% $CO_2$ in humidified atmosphere. All cell lines except for H2107 (NCI-H2107) and 293T were cultured in RPMI 1640 medium (Thermo Fisher Scientific, Waltham, MA, USA) supplemented with 10% fetal bovine serum (Thermo Fisher Scientific). H2107 cells were cultured in DMEM medium (Thermo Fisher Scientific) supplemented with 5% fetal bovine serum and 1% Glutamax (Thermo Fisher Scientific). 293 T cells were cultured in DMEM medium supplemented with 10% FBS. H1975 cells were cultured in stem cell culture media (Advanced DMEM/F12 [Thermo Fisher Scientific], 1% Glutamax, B-27 supplement [Thermo Fisher Scientific], 10 ng/mL recombinant human HB-EGF [PeproTech, Cranbury, NJ, USA], and 10 ng/mL recombinant human FGF-basic [PeproTech]) (*Park et al., 2018*) when indicated. H2107, H82 (NCI-H82), H524 (NCI-H524), H526 (NCI-H526), PC9 (PC-9), H1650 (NCI-H1650), H1975 (NCI-H1975), HCC827, HCC2279, HCC2935, HCC4006, HCC4011, H23 (NCI-H23), H1792, (NCI-H1792) A549, H358 (NCI-H358), H1395 (NCI-H1395), H2347 (NCI-H2347), H460 (NCI-H460), H1155 (NCI-H1155), and 293 T cells were obtained from American Type Tissue Culture (ATCC) or were a kind gift from Dr. Adi Gazdar (UTSW). Mycoplasma contamination check was carried out using a LookOut Mycoplasma PCR Detection Kit (Sigma-Aldrich, St. Louis, MO, USA). Cells were validated by STR profiling.

## Chemicals

Where indicated, the following chemicals were added to the media as indicated in the text: doxycycline hyclate (Sigma-Aldrich), SCH772984 (Selleck Chemicals, Houston, TX, USA), MK-2206 (Selleck Chemicals), RO4929097 (Selleck Chemicals), SB-747651A (Tocris Bioscience, Bristol, UK), A-485 (Tocris Bioscience), trichostatin A (Selleck Chemicals), C646 (Selleck Chemicals), ERGi-USU (Tocris Bioscience), and osimertinib (Selleck Chemicals).

## Microscopy

Fluorescence microscopy was performed using a digital inverted microscope AMF4300 (Thermo Fisher Scientific).

## Exploring mutation and copy number alteration data of KRAS and EGFR using cBioPortal

We collected the data of mutations and copy number alterations (amplification and deep deletion) of the *KRAS* and *EGFR* genes from 1314 lung cancer patients and 1316 samples using the cBio Cancer Genomics Portal (cBioPortal) database from the five studies as follows: TCGA, Firehose Legacy for adenocarcinoma (*N* = 586); TCGA, Firehose Legacy for squamouns cell carcinoma (*N* = 511); the Clinical Lung Cancer Genome Project (CLCGP) for SCLC (*Peifer et al., 2012*) (*N* = 29); Johns Hopkins University (JHU) Nat Genet 2012 for SCLC (*Rudin et al., 2012*) (*N* = 80); and U Cologne (UCOLOGNE) Nature 2015 for SCLC (*George et al., 2015*) (*N* = 110).

## Plasmids and generation of stable or transient cell lines

Plasmids used for expressing mutant KRAS (KRAS[G12V]), mutant EGFR (EGFR[L858R]), or GFP were identical to those described in our previous works (*Unni et al., 2018*; *Unni et al., 2015*). In brief, DNAs

encoding mutant KRAS, mutant EGFR, or GFP were cloned into pInducer20 that carries a tetracy-cline response element for doxycycline-dependent gene control and the tetracycline transactivator, rtTA, driven from the constitutive UbC promoter. Human *ETV1* (Addgene, Cambridge, MA, USA; plasmid #82209) and *ETV5* (Horizon Discovery, Cambridge, UK; clone 100008315) were transferred to pInducer20 using Gateway LR Clonase II enzyme mix (Thermo Fisher Scientific). Lentivirus was generated using the pInducer20-KRAS$^{G12V}$, -EGFR$^{L858R}$, -GFP, -ETV1, or-ETV5 as well as psPAX2 (Addgene; plasmid #12260) and pMD2.G (Addgene; plasmid #12259) in 293 T cells. After transduc-tion, stable polyclonal cells were selected with geneticin (Thermo Fisher Scientific) and single-cell-derived clonal cells were also established. Where indicated, doxycycline was added at the time of cell seeding at 100 ng/mL and cells were cultured for 72 hr and harvested unless otherwise stated. For 7-day time course experiments, medium was changed and doxycycline was refreshed on day 4. For other time course experiments, medium was changed and drugs were refreshed every 24 hr.

For transient overexpression of CIC-L, CIC-S, or CIC-S$^{V41G}$, three cell lines (H2107-KRAS$^{G12V}$, H82-KRAS$^{G12V}$, and H524-KRAS$^{G12V}$) were transfected with the corresponding constructs or with an empty pcDNA4/TO vector which had been kindly gifted from Dr. Wong (*Wong et al., 2019*) using Lipofectamine 2000 Reagent (Thermo Fisher Scientific). After 24 hr of transfection, medium was changed and doxycycline was added at 100 ng/mL. Cells were further cultured for 72 hr and harvested.

## CRISPR/Cas9 modification

The sgRNA sequences targeting *HES1* (sg*HES1*-1, 5'-GTGCTGGGGAAGTACCGAGC-3'; sg*HES1*-2, 5'-GGTATTAACGCCCTCGCACG-3'), *SOX9* (sg*SOX9*-1, 5'-CAAAGGCTACGACTGGACGC-3'; sg*SOX9*-2, 5'-AGGTGCTCAAAGGCTACGAC-3'), or *RB1* (5'-GCTCTGGGTCCTCCTCAGGA-3') were cloned into the lentiCRISPRv2 (Addgene #52961) plasmid. 293 T cells were transfected with recom-binant lentiCRISPRv2 together with psPAX2 and pMD2.G using Lipofectamine 2000 (Thermo Fisher Scientific). Undigested lentiCRISPRv2 plasmid lacking sgRNA sequence was used for pseudovirus production as a control. H2107-KRAS$^{G12V}$, H82-KRAS$^{G12V}$, and H524-KRAS$^{G12V}$ cells were infected with virus to knockout *HES1* or *SOX9*. PC9 and H1975 cells were infected with virus to knockout *RB1*. After maximally eliminating uninfected cells by selection with puromycin (Sigma-Aldrich), poly-clonal cells were collected. Single cell-derived clonal cells were also established after *HES1* or *RB1* knockout.

## Western blot analysis

Cells were lysed in RIPA Lysis and Extraction Buffer (G-Biosiences, St. Louis, MO, USA) containing Halt protease and phosphatase inhibitor cocktail (Thermo Fisher Scientific). For experiments of SCLC cell lines (H2107, H82, and H524) after mutant KRAS or mutant EGFR induction, both suspended and adherent cells were lysed and mixed together. Thus, lysates were representative of the entire population of cells. The only exception was lysates collected from adherent and non-adherent popu-lations separately as well as from both populations together to assess the heterogeneity of SCLC cells (*Figure 1—figure supplement 3b*). For experiments of acute and subacute treatment with osi-mertinib for up to 5 days, cells were serum starved for 24 hr and treated with osimertinib or 0.1% DMSO. Medium was changed and osimertinib was refreshed every 24 hr. Protein concentration was determined using a Pierce BCA protein assay kit (Thermo Fisher Scientific). 20 μg of lysates were denatured in NuPAGE LDS Sample Buffer (Thermo Fisher Scientific) and loaded on 4–12% Bis-Tris (Thermo Fisher Scientific) or 3–8% Tris-Acetate (Thermo Fisher Scientific) gradient gels. After electro-phoretic separation, the proteins were transferred onto PVDF membranes (MilliporeSigma, Billerica, MA, USA). The protein of interest was detected using an appropriate antibody specific for phospho-EGFR (Tyr1068) (2234; Cell Signaling Technology [CST], Danvers, MA, USA), EGFR (2232; CST), BRN2 (12137; CST), INSM1 (sc-271408; Santa Cruz Biotechnology, Dallas, TX), MASH1 (556604; BD Pharmingen Inc; San Diego, CA, USA), NEUROD1 (4373; CST), CD56 (3576; CST), SYP (4329; CST), phospho-ERK1/2 (Thr202/Tyr204; 9101; CST), ERK1/2 (4695; CST), phospho-AKT (Ser473; 4060; CST), AKT (4691; CST), RAS G12V mutant specific (14412; CST), RAS (8955; CST), MYC (5605; CST), cleaved PARP (5625; CST), YAP1 (14074; CST), phospho-RSK (Ser380; 11989; CST), RSK1/RSK2/ RSK3 (14813; CST), phospho-CREB (Ser133; 9196; CST), CREB (9197; CST), NOTCH1 (3608; CST), cleaved NOTCH1 (4147; CST), NOTCH2 (5732; CST), HES1 (11988; CST), SOX2 (3579; CST), SOX9

(82630; CST), H3K4me3 (9751; CST), H3K9me3 (13969; CST), H3K9ac (9649; CST), H3K14ac (7627; CST), H3K27ac (8173; CST), H3S10ph (9701; CST), H3S28ph (9713; CST), Histone H3 (4499; CST), CIC (PA1-46018; Thermo Fisher Scientific), HA-Tag (3724; CST), ETV1 (PA5-41484; Thermo Fisher Scientific), ETV4 (ab189826; Abcam), ETV5 (ab102010; Abcam), ERG (97249; CST), RB (9309; CST), β-Actin (12620; CST), or GAPDH (sc-47724; Santa Cruz Biotechnology) with ECL (Thermo Fisher Scientific).

## Cell proliferation assay

To determine the viability of cells over a 5-, 7-, or 8-day time course for H82, H524, or H2107 derivatives, respectively, cells with doxycycline-inducible constructs were seeded in triplicate in 6-well plates with (100 ng/mL) or without doxycycline at $8.0 \times 10^4$ (H2107 derivatives), $1.5 \times 10^4$ (H82 derivatives), and $4.0 \times 10^4$ (H524 derivatives) cells/well. Medium was not changed during the experiments. At indicated time points, an alamarBlue cell viability agent (Thermo Fisher Scientific) was added and intensities were measured for each well using a Cytation 3 Multi Modal Reader with Gen5 software (BioTek Instruments, Inc, Winooski, VT, USA). Along with cell viability, cell numbers were also counted at indicated time points in triplicate.

## Soft agar colony formation assay

Soft agar colony formation assay was performed using a CytoSelect 96-Well Cell Transformation Assay Kit (Cell BioLabs, Inc, San Diego, CA) following the manufacturer's protocol. In brief, 0.6% CytoSelect agar was added to the bottom of a 96-well plate prior to seeding each well with 10,000 cells suspended in 0.3% CytoSelect agar. After the agar had solidified, cells were then treated with either with media control or media supplemented with 100 ng/mL doxycycline. Cells were cultured for 5 days prior to solubilizing the agar, lysing the cells, and subsequent quantification by CyQuant GR Dye (fluorescence: 485/520 nm).

## Assessment of phenotypic change from a suspended to adherent state

To determine the ability of phenotypic change in the growing pattern from a suspended to adherent state, cells with doxycycline-inducible constructs were seeded in triplicate in 6-well plates with (100 ng/mL) or without doxycycline at $1.7 \times 10^6$ (H2107 derivatives), $4.0 \times 10^4$ (H82 derivatives), and $1.0 \times 10^6$ (H524 derivatives) cells/well. After incubation of cells for 7 days (H2107 and H82 derivatives) or 5 days (H524 derivatives) without medium change, medium containing suspended cells was removed and adherent cells were washed with PBS and then medium was replaced. Viability of adherent cells was assessed using an alamarBlue cell viability agent. Adherent cells were also fixed and stained with crystal violet.

To assess the impact of SCH772984 (1 μM) and/or MK-2206 (10 μM), or SB-747651A (5 μM) and/or MK-2206 (10 μM) on the phenotypic change from a suspended to adherent state, doxycycline-inducible KRAS$^{G12V}$ cells were seeded in triplicate in six-well plates with (100 ng/mL) or without doxycycline and with or without indicated drugs at $2.0 \times 10^6$ cells/well. After incubation for 72 hr, medium containing cells in suspension was aspirated and then medium containing indicated doxycycline and/or drugs was replaced. Cell viability of adherent cells was evaluated using an alamarBlue cell viability agent. Adherent cells were also fixed and stained with crystal violet.

## Gene expression profiling and gene set enrichment analysis

Total RNA was extracted in triplicate using a *Quick*-RNA Miniprep Kit (Zymo Research, Irvine, CA, USA) according to the manufacturer's protocol from mutant KRAS-, mutant EGFR-, or GFP-transduced stable H2107 and H82 cells on doxycycline treatment day 1 and day seven as well as from non-doxycycline-treated control cells. Sample quality was assessed using an Agilent Bioanalyzer (Agilent, Santa Clara, CA) and subsequent sample preparation, array hybridization, and data acquisition was performed by the Centre for Applied Genomics Microarray Facility (Toronto, Ontario). The GeneChip Human Gene 2.0 ST Assay (Thermo Fisher Scientific) was used according to the manufacturer's protocols. Raw data were normalized by robust multiarray analysis via the RMA package (*Irizarry et al., 2003*) and subsequently analyzed to detect genes differentially expressed between EGFR$^{L858R}$- vs GFP-expressing cells and KRAS$^{G12V}$- vs GFP-expressing cells at each time point for each cell line using a generalized linear regression model and applying an empirical Bayesian fit

through the limma package (*Ritchie et al., 2015*) in R (R Foundation for Statistical Computing, Vienna, Austria, version 3.6.1). Differentially expressed genes in EGFR[L858R] or KRAS[G12V] vs GFP at each time point with Benjamini–Hochberg corrected *P* values < 0.05 were considered significant. Significantly upregulated or downregulated genes in KRAS[G12V]-induced cells over GFP controls were analyzed by Enrichr (*Chen et al., 2013*; *Kuleshov et al., 2016*) separately to identify enriched ENCODE and ChEA consensus TFs from the ChIP-X database. Gene Set Enrichment Analysis (GSEA) was performed using GSEA software version 4.0.3 with default parameters using the gene set obtained from hallmark gene sets (*Subramanian et al., 2005*). Gene expression data has been deposited in the Gene Expression Omnibus (GEO, accession number GSE160482). Additional micro-array data for a panel of LUAD and SCLC cell lines was downloaded from GEO (GSE4824) and plotted using Morpheus (https://software.broadinstitute.org/morpheus) as previous described (*Lockwood et al., 2008*).

## Phospho-kinase array analysis

The Proteome Profiler Human Phospho-Kinase Array Kit (R and D Systems, Minneapolis, MN, USA) was purchased and phosphorylation profiles of kinases were analyzed according to the manufacturer's protocol.

## Reverse transcription and quantitative real-time PCR analysis

Total RNA was isolated from cell lines as described above and was reverse transcribed to cDNA using a High-Capacity RNA-to-cDNA Kit (Thermo Fisher Scientific). Real-time quantitative PCR reactions were performed using TaqMan Gene Expression Assay Mix and TaqMan Universal PCR Master Mix (Thermo Fisher Scientific) with the 7500 Fast Real Time PCR System (Thermo Fisher Scientific). The TaqMan Gene Expression Assay Mix for *REST* (Hs05028212_s1) was obtained from Thermo Fisher Scientific. The ΔΔCt method was used for relative expression quantification using the average cycle thresholds. The relative expression of *REST* represents an average of triplicates that are normalized to the transcription levels of beta-actin (Hs99999903_m1; Thermo Fisher Scientific).

## RNA interference

Approximately $1.5 \times 10^6$ cells were transfected with ON-TARGETplus siRNA pools (Horizon Discovery) using DharmaFECT one transfection reagent (Horizon Discovery) at a final concentration of 50 nM against the following targets: *MAPK3* (L-003592–00), *MAPK1* (L-003555–00), *REST* (L-006466–00), *CIC* (L-015185–01), *ETV4* (L-004207–00), and *ETV5* (L-008894–00) as well as a non-targeting control (D-001810–10). Cells were cultured for 72 hr after transfection and used for further analyses. Where indicated, doxycycline was added at 100 ng/mL at the time of transfection.

## Dose-response analysis

Cells of PC9 and H1975 derivatives were seeded in 96-well plates at a density of $1.5 \times 10^3$ cells per well. After 24 hr, osimertinib was added at different concentrations. Cells were allowed to grow for 72 hr after osimertinib addition and cell viability was assessed using alamarBlue cell viability agent.

## Generation of osimertinib-resistant cells

To generate resistant cell lines to osimertinib, we exposed *RB1*-proficient or -deficient PC9 and H1975 cells to the drug by either stepwise dose-escalation (starting at 10 nM or 30 nM and ending with 1 µM or 2 µM for PC9 and H1975, respectively) or initial high-dose (1 µM) method. Osimertinib was refreshed every 3 or 4 days. To capture possible SCLC-transformed cells which were anticipated to be likely in suspension, we passaged both adherent and suspended cells together during making cells resistant to osimertinib. Resistant cells were maintained as polyclonal populations under constant exposure to the drugs.

## Analysis of acquired genomic alterations by MSK-IMPACT

Cell lines were profiled by the MSK-IMPACT (Integrated Mutation Profiling of Actionable Cancer Targets) platform which is a hybridization capture-based next generation sequencing (NGS) assay for targeted deep sequencing of exons and selected introns of 468 cancer-associated genes and gene fusions (*Cheng et al., 2015*). The assay detects mutations and copy-number alterations. Genomic

DNA was extracted from osimertinib-resistant cells as well as matched parental cells using a DNeasy Blood and Tissue Kit (Qiagen, Hilden, Germany). We reviewed all candidate alterations identified in resistant cells as well as parental cells and considered those identified only in resistant cells as candidate acquired resistance genomic alterations to osimertinib.

## Subcellular fractionation

H2107, H82, and H524 cell lines were subjected to subcellular fractionation using NE-PER Nuclear and Cytoplasmic Extraction Reagents (Thermo Fisher Scientific) according to the manufacturer's instruction. Fractionation efficiency was confirmed by western blot analysis using MYC as nuclear and GAPDH as cytoplasmic protein controls, respectively.

## ATAC-seq analysis

H2107-KRAS$^{G12V}$ and H524-KRAS$^{G12V}$ cells were treated with doxycycline ±SCH772984 (1 µM) for 72 hr. H82-KRAS$^{G12V}$ cells were treated with the following chemicals: doxycycline; doxycycline +-SCH772984 (1 µM); doxycycline +SB-747651A (5 µM); doxycycline +A-485 (400 nM); or doxycycline +SB-747651A (5 µM)+A-485 (400 nM). After 72 hr treatment, these cells as well as corresponding non-treated control cells were collected and frozen. ATAC-seq was performed using the Omni-ATAC protocol (*Corces et al., 2017*) with slight modifications as below. In brief, cells were resuspended in nuclear lysis buffer (10 mM Tris-HCl pH 7.4; 10 mM NaCl; 3 mM MgCl2; 0.1% NP-40 0.1% Tween-20, 0.01% Digitonin) on ice, then spun down in a cold centrifuge at 600 x g for 10 min, resuspended in RSB Tween and nuclei were quantified using Trypan Blue (Invitrogen) on a Countess II Counter (Invitrogen). An aliquot of 50,000 nuclei per sample was transferred to a fresh tube, spun down, resuspended in transposition solution and transposed for 30 min at room temperature as described previously (*Corces et al., 2017*). Libraries were prepared using standard Illumina Nextera indices. Library cleanup and dual-sided size selection was performed using SPRIselect beads (Beckman Coulter) with 0.4X and 1.2X ratios. Sequencing was performed on a NextSeq 500 (Illumina) with 150 cycles on a high-output cartridge in paired-end mode at the Center for Genomics and Health Informatics (CHGI) at the Cumming School of Medicine (University of Calgary). On average, 73,950,218 reads were generated for each library (range: 50,913,762–94,506,244 reads). Data has been deposited in GEO (GSE160204).

Sequencing data were aligned using bwa (0.7.17) to the hg38 assembly of the human genome (*Li and Durbin, 2009*). Extraneous chromosomes and low-quality reads were removed using SAMtools (v 1.10) (*Li et al., 2009*) and PCR duplicates were removed using Picard tools (Broad Institute). Peaks were called using MACS2 (*Zhang et al., 2008*) using the following parameters: -g hs -q 0.05 – `shift` −100 `−extsize` 200 `−nomodel` -B `−keep-dup all`, followed by pileup construction and fold-change graph generation using macs2 bdgcmp. A union peaklist across all samples was generated using BEDTools (*Quinlan and Hall, 2010*), and absolute signal at each peak was extracted from each sample. These counts tables were analyzed using DESeq2 (*Love et al., 2014*) in R to identify differentially accessible regions, with the following cut-offs: absolute log fold change greater than 1.5, p<0.01, and minimum peak signal of 20,000. Motif analysis of differentially accessible regions was performed using the findMotifsGenome function of HOMER (v4.11) (*Heinz et al., 2010*). Motif profiles were generated using HOMER. Motif enrichment rankings were computed using a method described previously (*Park et al., 2018*). In brief, for each condition, motif enrichment lists in regions of lost and increased accessibility were arranged by fold change and p value, assigning each a separate rank for regions of lost and increased accessibility. Motif ranks in regions of lost and increased accessibility were averaged over all samples. Final score was obtained by subtracting the rank order of each motif in the increased accessibility regions from rank order in the regions of lost accessibility, and motifs were arranged in descending order by score. Permutation analysis was conducted using regioneR (*Gel et al., 2016*), with 500 permutations, using publicly available H3K27ac data for human lung from the Roadmap Epigenomics consortium (*Bernstein et al., 2010*) (GEO ID: GSM906395).

## Statistical analysis

Differences in continuous variables were analyzed by the Student's *t* tests or one-way ANOVA followed by the Holm's multiple comparisons post-test. IC$_{50}$ values in dose-response analyses were compared by the extra sum-of-squares F test. The statistical analyses were performed using R

software, version 3.6.1 and GraphPad Prism, version 8.2.1 (GraphPad Software, San Diego, CA, USA). All statistical tests were two-sided. p Values < 0.05 were considered statistically significant. Data are presented as mean ± SEM or mean ± SD of a minimum of three independent experiments.

## Acknowledgements

This work was funded by the Canadian Institutes of Health Research (CIHR; PJT-148725), the British Columbia Lung Association (Research Grant) and the Terry Fox Research Institute (New Investigator Award) to WWL; a Canada Research Chair in Brain Cancer Epigenomics (tier 2) from the Government of Canada and a Project grant from CIHR (PJT-156278) to MG; a Clinician Investigator Program fellowship from Alberta Health Services and a fellowship from Alberta Innovates to AN; and a Lilly Oncology Fellowship Program Award from the Japanese Respiratory Society and a fellowship from the Michael Smith Foundation for Health Research (MSFHR) to YI. WWL. is a MSFHR Scholar and CIHR New Investigator.

## Additional information

### Funding

| Funder | Grant reference number | Author |
|---|---|---|
| Canadian Institutes of Health Research | PJT-148725 | William W Lockwood |
| British Columbia Lung Association | Research Grant | Yusuke Inoue William W Lockwood |
| Terry Fox Research Institute | New Investigator Award | William W Lockwood |
| Canadian Institutes of Health Research | PJT-156278 | Marco Gallo |
| Canada Research Chairs | Brain Cancer Epigenomics (Tier 2) | Marco Gallo |
| Alberta Health Services | Clinician Investigator Program fellowship | Ana Nikolic |
| Alberta Innovates | Fellowship | Ana Nikolic |
| Japanese Respiratory Society | Lilly Oncology Fellowship Program Award | Yusuke Inoue |
| Michael Smith Foundation for Health Research | Fellowship | Yusuke Inoue |
| Michael Smith Foundation for Health Research | Scholar Award | William W Lockwood |
| Canadian Institutes of Health Research | New Investigator Award | William W Lockwood |

The funders had no role in study design, data collection and interpretation, or the decision to submit the work for publication.

### Author contributions

Yusuke Inoue, Conceptualization, Data curation, Formal analysis, Funding acquisition, Validation, Investigation, Visualization, Methodology, Writing - original draft, Project administration, Writing - review and editing; Ana Nikolic, Data curation, Formal analysis, Validation, Investigation, Visualization, Methodology, Writing - original draft; Dylan Farnsworth, Formal analysis, Investigation, Methodology; Rocky Shi, Fraser D Johnson, Investigation; Alvin Liu, Data curation, Investigation, Visualization, Methodology; Marc Ladanyi, Data curation, Supervision, Methodology; Romel Somwar, Data curation, Formal analysis, Investigation; Marco Gallo, Conceptualization, Resources, Data curation, Formal analysis, Supervision, Funding acquisition, Investigation, Methodology, Writing - original draft, Writing - review and editing; William W Lockwood, Conceptualization, Resources, Formal

analysis, Supervision, Funding acquisition, Validation, Investigation, Methodology, Writing - original draft, Project administration, Writing - review and editing

### Author ORCIDs
Yusuke Inoue (ID) https://orcid.org/0000-0001-8075-0597
Dylan Farnsworth (ID) http://orcid.org/0000-0002-2402-159X
William W Lockwood (ID) https://orcid.org/0000-0001-9831-3408

### Decision letter and Author response
Decision letter https://doi.org/10.7554/eLife.66524.sa1
Author response https://doi.org/10.7554/eLife.66524.sa2

## Additional files

### Supplementary files
• Supplementary file 1. Genes differentially expressed in H82 and H2107 cells after induction of KRAS$^{G12V}$ or EGFR$^{L858R}$ for 1 or 7 days.

• Transparent reporting form

### Data availability
Gene expression data has been deposited to GEO under the accession code GSE160482. ATAC seq data has been deposited to GEO under the accession code GSE160204.

The following datasets were generated:

| Author(s) | Year | Dataset title | Dataset URL | Database and Identifier |
|---|---|---|---|---|
| Inoue Y, Nikolic A, Liu A, Ladanyi M, Somwar R, Gallo M, Lockwood WW | 2020 | ATAC-seq of lung cancer cell lines with doxycycline KRAS G12V and drug treatments | https://www.ncbi.nlm.nih.gov/geo/query/acc.cgi?acc=GSE160204 | NCBI Gene Expression Omnibus, GSE160204 |
| Lockwood WW, Inoue Y | 2020 | Response of SCLC to mutant KRAS or EGFR induction | https://www.ncbi.nlm.nih.gov/geo/query/acc.cgi?acc=GSE160482 | NCBI Gene Expression Omnibus, GSE160482 |

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
