## [Decision Letter]

**Acceptance summary:**

Inoue and colleagues show that RAS-driven ERK signaling suppresses neuro-endocrine differentiation in lung cancer via a mechanism that requires ERK2 and CBP/p300-driven chromatin changes. This is consistent with the frequent loss in small cell lung carcinoma of CBP/p300. The authors show that Ras-Erk signaling induces multiple ETV family members in an ERK-dependent manner. They provide data that ETVs can suppress neuro-endocrine differentiation, and that knockdown of ETVs can provide rescue of Ras/Erk-mediated neuro-endocrine suppression. This manuscript will be of interest to cancer biologists studying cell fate transitions, particularly adenocarcinoma-to-small cell transitions that occur in prostate and lung cancer.

**Decision letter after peer review:**

Thank you for submitting your article "Extracellular signal-regulated kinase mediates chromatin rewiring and lineage transformation in lung cancer" for consideration by *eLife*. Your article has been reviewed by 3 peer reviewers, and the evaluation has been overseen by Maureen Murphy as the Senior and Reviewing Editor. The following individuals involved in review of your submission have agreed to reveal their identity: Igor Astsaturov (Reviewer #1); Trudy Oliver (Reviewer #2); John Minna (Reviewer #3).

Essential revisions:

The reviewers agree that this is a copious amount of careful work that will be important for the field. In addition, this appears to be an important new area for SCLC work, with potential therapeutic implications. Addressing as many as possible of the major concerns below is encouraged; it is at the authors discretion as to which concerns they will choose to address experimentally, and which they choose to address in the text. A revised manuscript that addresses at least some of the concerns below experimentally is anticipated, and might be expected to fare well upon re-review.

1) The manuscript is densely, and at times confusingly, written. The authors need to edit their manuscript to make it easier to read and understandable for non-experts in the field. While they have some summary figures, I think the manuscript, in terms of editing, would benefit from having a main summary figure, or discussion that clearly depicts the overarching conclusions from the experiments.

2) The paper provides no conclusive evidence that epithelial -to-NE transdifferentiation is possible to model in vitro. Even the opposite NE-to-epithelial trans-differentiation is only partial. The authors are encouraged to provide data which demonstrate the existence of dynamic switch from the epithelial to neuroendocrine differentiation based on the model proposed; the idea that withdrawal of upstream signaling and shutting down the MAPK pathway leads to de-repression of NE TFs should be illustrated. One model, for example, is the iKRAS mouse model, either Tet-ON or Tet-OFF. There are also many cell lines available derived from these tumors. The authors could potentially pull back KRAS expression and show acquired NE features. Other alternatives may include using DN-MEK to address this issue. In addition, the authors do not really demonstrate that this mechanism takes place in vivo. For example, the authors could use Tet-regulated KRAS model to illustrate a morphological switch to adenocarcinoma from SCLC upon KRAS upregulation.

3) Despite zeroing in on ETVs downstream of ERK1/2, the paper does not go as far as showing the direct effect of these TFs as repressors of NE differentiation (ASCL1, BRN2, NEUROD1 etc.). The authors should be encouraged to demonstrate an association of ETS-high profile with adenocarcinoma and a negative association with SCLC. Can the authors address the question, would overexpression of ETV (ETV5 in one of their cell lines) produce similar results to KRAS overexpression?

4) The authors are encouraged to add some biologically relevant data regarding therapeutics (that is, data that MAPK ERK activation inhibits the in vitro or in vivo growth of SCLC), for example as xenografts or in colony formation assays. For example, what are the effects of EGFR, KRAS oncogene activation in vivo in xenografts or in colony formation assays?

5) There is an excessive amount of data and some of the negative data detracts from the focus of the story. I would suggest the authors remove some figures that are either irrelevant to the key points and are not further validated like Supp Figures S5, S6, S7, or are simply confirmatory of what should be known such as Supp Figure S17. S9 could easily be combined with S8. Lines 279-286 discuss "data not shown" related to HEY1 that could easily be omitted.

6) One recurring issue in the manuscript is that the observations are often not consistent across the three cell lines and are context-specific effects, and the potential reasons could be better explained. The cell lines chosen unfortunately (but interestingly) represent some of the major cell states of SCLC. H2107 represents the ASCL1+ NE-high subset of SCLC (and has some MYCL). H82 and H524 represent the C-Myc (MYC)-high subset of SCLC, with H82 having a MYC amplification, and both representing the NEUROD1 subtype (which tend to be associated with more MYC). Assessment of NE score using a common approach in the field (Zhang et al., TLCR) shows that H82 cells are already considerably NE-low, with H524 as NE-intermediate/high, and H2017 as NE-high. So, this may be related to why H82 seemed to be the most permissive cell line to change NE fate in multiple assays. In addition, H2107 and H524 appear to have EP300 mutations, which may contribute to their NE-high nature and contribute to the refractory response to A485 treatment based on the author's model. It's known that MYCL and MYC-driven cell lines differ in numerous aspects from transcriptional signatures, super enhancer usage, metabolic regulation, therapeutic response, etc. This information could be mentioned in the results and discussed when mentioned as a factor near line 540.

7) Related to Figure 4, the authors show that p300 pharmacological inhibition can restore NE fate in presence of Kras. Given that drugs can have off-target effects, it would be helpful to know if genetic knockdown/knockout of p300 phenocopies these effects. Given that CREBBP (CBP) or EP300 (p300) mutations are common in SCLC, it is also relevant whether any of these cell lines have CREBBP (CBP) or EP300 (p300) mutations. It appears H2107 and H524 may have EP300 mutations. Have they tried to restore EP300 function?

8) Related to Figure 6, did the authors test whether ETV4 is sufficient to reduce NE markers? Given that multiple ETV family members can repress NE markers, it seems plausible that it might take knockdown of all of them to rescue Kras-induced NE suppression. This could be understandably technically difficult, so perhaps the authors could just make this point clear in the text.

9) Related to FiguresS8/S10, did the authors look at REST protein levels in S8 and/or S10? The functional studies for REST knockdown appear performed in only 1 of the 3 cell lines?

10) Related to Figure S18e and S18g: This figure is confusing and could better explained in the text. In S18e, the mutation status of these genes in parental/sgControl cells prior to Osimertinib treatment should be included. Mutation status appears to be presented only after two different types of Osimertinib exposure. Second, in S18g, sgControl resistant cell lose expression of CIC protein expression entirely in the stepwise and initial exposure conditions. Consistently, sgControl Resistant cells seem to increase ETV4 and ETV5 protein levels compared to sgControl parental cells. The authors could explain whether this is due to the gain of the specific nonsense mutation. Since gain of ETV4/5 presumably inhibits NE fate conversion; does ETV knockout induce NE fate in this context?

11) The authors may need to address certain aspects of the principal observation that the MAPK blockade resistant cells acquire NE features. Specifically, these differentiation switches typically occur in complex multiclonal populations of cancer cells in vivo, not in established cell lines. Using human PDX models or mouse models of KRAS/EGFR-driven lung cancers, the authors could have the opportunity to demonstrate the existence of such clonal shifts. Starting with in vitro established stable NE or epithelial clones could be an exceedingly difficult model in which to observe these differentiation switches. Can the authors address this point in the Discussion.

[Editors' note: further revisions were suggested prior to acceptance, as described below.]

Thank you for submitting your article "Extracellular signal-regulated kinase mediates chromatin rewiring and lineage transformation in lung cancer" for consideration by *eLife*. Your article has been reviewed by 3 peer reviewers, and the evaluation has been overseen by Maureen Murphy as the Senior and Reviewing Editor. The following individuals involved in review of your submission have agreed to reveal their identity: Igor Astsaturov (Reviewer #1); Trudy Oliver (Reviewer #2); John Minna (Reviewer #3).

The reviewers have discussed their reviews with one another, and the Reviewing Editor has drafted this to help you prepare a revised submission. A few revisions are requested for clarity; these are detailed below.

*Reviewer #1:*

The authors largely addressed the critical points raised in the review process. While the study was designed to determine the effect of MAPK pathway activation on NE differentiation, a more clinically relevant epithelial-to-NE switch upon MAPK blockade with TKI in lung adenocarcinoma has not been modeled here, which is the limitation of this paper. With this limitation in mind, the presented work has shown several important regulatory mechanisms, namely, that hyperactive ERK signaling is a default repressor of neuroendocrine differentiation transcriptional regulators. ERK2 appears to be the dominant kinase in this process, and the downstream-induced ETV family TFs are important in executing this mechanism.

1. Certain statements in the text of the paper need to be made more consistent. For example, in line 252 and below, the authors state that both, AKT and ERK are important "for the phenotypic transition", whereas the preceding discussion (line 225) states the opposite.

2. In the results and discussion, the authors should explicitly state that long-term induction with Doxy of KRAS has been toxic as early as day 7. Longer cultures (day 28) showed complete loss of KRAS suggesting a negative selection.

3. Since many Western blot images are not statistically processed (which would be a reasonable request to the authors), explain if the figure legends whether the presented changes are representative of a certain number of repeats. It remains unclear how robust and reproducible are the results at the protein level. Certainly, concordant changes in transcriptional signatures and Westerns is a plus.

4. Figure 1e presents "viability of adherent cells": please explain or correct the label to reflect exactly what was measured. If this is colony or adherent cell density based on the colony formation, then it should be stated. Unclear if the non-adherent population was simply removed in the process of staining.

5. As relevant to most Western blot images, the Method section should state if an effort was made to retain the non-adherent population during cell collection for lysates, and if the presented lysates are representative of the total or adherent population only. This technical detail is important to interpret the results since the authors noticed growth pattern changes with introduction of MAPK-activating oncogenes.

*Reviewer #2:*

Overall, Inoue et al. have significantly improved the manuscript by consolidating and removing some figures, and making the data more organized and concise. The Discussion is also significantly improved by detailing the caveats and considerations of the study.

*Reviewer #3:*

The authors have responded appropriately to all of the reviewers' comments including extensive editing and providing additional new data.

---

## [Author Response]

Essential revisions:The reviewers agree that this is a copious amount of careful work that will be important for the field. In addition, this appears to be an important new area for SCLC work, with potential therapeutic implications. Addressing as many as possible of the major concerns below is encouraged; it is at the authors discretion as to which concerns they will choose to address experimentally, and which they choose to address in the text. A revised manuscript that addresses at least some of the concerns below experimentally is anticipated, and might be expected to fare well upon re-review.1) The manuscript is densely, and at times confusingly, written. The authors need to edit their manuscript to make it easier to read and understandable for non-experts in the field. While they have some summary figures, I think the manuscript, in terms of editing, would benefit from having a main summary figure, or discussion that clearly depicts the overarching conclusions from the experiments.

We appreciate the reviewers highlighting the dense nature of our manuscript and that this may convolute our overall message, especially to non-experts in the field. We have substantially edited the manuscript (see attached version with changes tracked) to streamline the Results sections and revamped the discussion to better articulate our main conclusions. We feel this now better reflects the major findings from our work.

2) The paper provides no conclusive evidence that epithelial -to-NE transdifferentiation is possible to model in vitro. Even the opposite NE-to-epithelial trans-differentiation is only partial. The authors are encouraged to provide data which demonstrate the existence of dynamic switch from the epithelial to neuroendocrine differentiation based on the model proposed; the idea that withdrawal of upstream signaling and shutting down the MAPK pathway leads to de-repression of NE TFs should be illustrated. One model, for example, is the iKRAS mouse model, either Tet-ON or Tet-OFF. There are also many cell lines available derived from these tumors. The authors could potentially pull back KRAS expression and show acquired NE features. Other alternatives may include using DN-MEK to address this issue. In addition, the authors do not really demonstrate that this mechanism takes place in vivo. For example, the authors could use Tet-regulated KRAS model to illustrate a morphological switch to adenocarcinoma from SCLC upon KRAS upregulation.

The reviewers are correct that we were unable to conclusively model the epithelial to NE transdifferentiation process that underlies EGFR TKI resistance in lung adenocarcinoma. As I am sure the reviewers appreciate, despite great effort in the lung cancer research community, this has been a very difficult process to model to date, either in vitro or in vivo. While our initial intent was to use the information from the NE to epithelial switch to “reverse engineer” the reciprocal process in *EGFR*-mutant lung adenocarcinoma, the combination of molecular changes and treatments we employed, in addition to specific culture conditions, was insufficient to initiate this process. We argue that this highlights the complex nature of this transformation, and suggests that multiple factors, potentially occurring in a specific order, are required for LUAD to SCLC transformation to occur. However, we feel that our findings regarding the NE to epithelial transformation process will be essential for modeling this process in the future as it indicates a key role for ERK and CBP/p300 suppression in mediating this shift, along with *TP53* and *RB1* mutation. The multifactorial nature of this process is also why we have focused on cell models for the current studies. While the experiments that the reviewers propose, specifically inducible mutant *KRAS* mouse models of LUAD, have the potential to investigate the impact of ERK signaling on NE differentiation, our results suggest that they would likely require additional genetic alterations, including *TP53* and *RB1* knockout and CBP (*CREBBP*)/p300 (*EP300*) mutation, for a phenotype to be observed. However, generating mouse model systems with such a background (TetO-Mutant-KRAS, p53 knockout, RB knockout, CBP/p300 knockout) is unfeasible for the timeline of this manuscript, and the experiments with these systems will likely take many months to conduct. For example, our in house TetO-KRAS-G12D mouse models take upwards of 10 months to establish LUAD tumors. Our future plans will employ these models as well as TetO-Mutant-EGFR mice combined with clinically relevant treatments (osimertinib etc) and genetic alterations based on our current work to explore this concept in greater detail.

Along with issues related to clonality and heterogeneity (Comment 12) we have now stated the potential limitations of our focus on established cancer cell lines for this work and described experimental systems that may provide useful for future studies in the Discussion section. However, we feel that despite our inability to achieve LUAD to SCLC transformation, our work still provides extremely valuable information regarding the factors that control cell state transformations in lung cancer that will be useful to the research community.

3) Despite zeroing in on ETVs downstream of ERK1/2, the paper does not go as far as showing the direct effect of these TFs as repressors of NE differentiation (ASCL1, BRN2, NEUROD1 etc.). The authors should be encouraged to demonstrate an association of ETS-high profile with adenocarcinoma and a negative association with SCLC. Can the authors address the question, would overexpression of ETV (ETV5 in one of their cell lines) produce similar results to KRAS overexpression?

We appreciate the reviewers’ encouragement to further define the association between ETVs and NE differentiation and have added substantial experimental data to address this as part of the revised Figure 6. First, using expression data for a panel of lung adenocarcinoma and SCLC cell lines, we demonstrate the exclusivity between ETV1/4/5 expression – which are upregulated in adenocarcinomas – and NE transcription factors (ASCL1, BRN2, NEUROD1 and INSM1) – which are up in SCLC – as suggested (Figure 6B). Furthermore, we have now conditionally expressed ETV1 in all three SCLC cell lines (H82, H2107 and H524) and ETV5 in two SCLC cell lines (H82 and H524) which both induced the downregulation of NE transcription factors, similar to the results with forced mutant KRAS expression (Figures 6D and 6E). Together, we feel this additional work clearly demonstrates the association between ETV expression and suppression of the NE differentiation program, further defining these factors as primary downstream drivers of ERK in this context.

4) The authors are encouraged to add some biologically relevant data regarding therapeutics (that is, data that MAPK ERK activation inhibits the in vitro or in vivo growth of SCLC), for example as xenografts or in colony formation assays. For example, what are the effects of EGFR, KRAS oncogene activation in vivo in xenografts or in colony formation assays?

The reviewers bring up an interesting point regarding the therapeutic potential of our findings for SCLC treatment. While we do demonstrate that activation of mutant KRAS has inhibitory effects in some of the cell lines tested, namely the ASCL1+ H2107 cell line (Figure 1—figure supplement 2b and c), it also had a stimulatory effect on the NEUROD1+ H82 cell line, despite the downregulation of NE differentiation. Thus, it is possible that there are distinct SCLC contexts in which ERK activation may be lethal, and this warrants further investigation, especially given our previous observation that hyperactivation of ERK is also lethal in NSCLC (Unni et al. *eLife* 2018). However, despite differences in regard to cell growth, the transdifferentiation phenotype upon ERK induction was consistent across all SCLC cell lines, which is why we focused on this aspect for the study. Given the reviewers have already suggested that the manuscript is very dense, and contains copious amounts of data, we feel that further exploring the therapeutic potential of ERK induction in a larger panel of SCLC cells is better suited as a focus for a follow up study.

However, as suggested by the reviewers, we have now added soft agar colony formation assay data for all three SCLC cell lines with activated KRAS, demonstrating similar results as those seen in proliferation assays. This is now presented as Figure 1—figure supplement 2d.

5) There is an excessive amount of data and some of the negative data detracts from the focus of the story. I would suggest the authors remove some figures that are either irrelevant to the key points and are not further validated like Supp Figures S5, S6, S7, or are simply confirmatory of what should be known such as Supp Figure S17. S9 could easily be combined with S8. Lines 279-286 discuss "data not shown" related to HEY1 that could easily be omitted.

We agree with the reviewers that some of the supplemental data was excessive and likely not needed to assert the conclusions of our study. As suggested, we have now revised the Supplemental Figures (including renaming in accordance with e*Life* guidelines), removing original Supplemental Figures S5 and S17 as well as panels from original Supplemental Figure S13. In addition, the original Supplemental Figure S9 was combined with Supplementary Figure S8 as suggested by the reviewers. Finally, we have minimized or removed much of the results related to negative data, mainly the data and section in reference HEY1 that were highlighted as a distraction from the reviewers. Together, we feel these changes have streamlined the manuscript, allowing the reader to better follow the main premise and findings of the work.

6) One recurring issue in the manuscript is that the observations are often not consistent across the three cell lines and are context-specific effects, and the potential reasons could be better explained. The cell lines chosen unfortunately (but interestingly) represent some of the major cell states of SCLC. H2107 represents the ASCL1+ NE-high subset of SCLC (and has some MYCL). H82 and H524 represent the C-Myc (MYC)-high subset of SCLC, with H82 having a MYC amplification, and both representing the NEUROD1 subtype (which tend to be associated with more MYC). Assessment of NE score using a common approach in the field (Zhang et al., TLCR) shows that H82 cells are already considerably NE-low, with H524 as NE-intermediate/high, and H2017 as NE-high. So, this may be related to why H82 seemed to be the most permissive cell line to change NE fate in multiple assays. In addition, H2107 and H524 appear to have EP300 mutations, which may contribute to their NE-high nature and contribute to the refractory response to A485 treatment based on the author's model. It's known that MYCL and MYC-driven cell lines differ in numerous aspects from transcriptional signatures, super enhancer usage, metabolic regulation, therapeutic response, etc. This information could be mentioned in the results and discussed when mentioned as a factor near line 540.

We appreciate the reviewers pointing out this important consideration. As mentioned above (Response 4), we also see a variable growth phenotype upon KRAS induction depending on SCLC subtype. We have now addressed this important information, including the *EP300* mutation status, in results and in the discussion – with appropriate citations – where indicated by the reviewers.

7) Related to Figure 4, the authors show that p300 pharmacological inhibition can restore NE fate in presence of Kras. Given that drugs can have off-target effects, it would be helpful to know if genetic knockdown/knockout of p300 phenocopies these effects. Given that CREBBP (CBP) or EP300 (p300) mutations are common in SCLC, it is also relevant whether any of these cell lines have CREBBP (CBP) or EP300 (p300) mutations. It appears H2107 and H524 may have EP300 mutations. Have they tried to restore EP300 function?

The reviewers bring up an interesting experiment we had previously considered. Once identifying H3K27ac as a major change upon KRAS/ERK induction in SCLC, we originally considered genetic knockout to determine whether it could rescue transdifferentiation. However, using a different compound, C646, which only inhibits p300, we saw modest rescue of NE factor suppression upon KRAS induction, suggesting that dual p300/CBP inhibition was necessary (Figure 4—figure supplement 1). For this reason, we decided to use A-485, which inhibits both p300 and CBP, as opposed to a genetic approach of dual knockdown which would prove more complicated to achieve.

As mentioned above in Response 6, we have now clearly indicated the mutation status of *EP300* in the SCLC cell lines used in the study, and the potential influence these mutations could have on the results observed. Restoring *EP300* function in these mutant SCLC cell lines is a very interesting experiment that would be predicted to increase H3K27ac levels mimicking the effects of ERK induction, leading to the downregulation of NE factors. While we have not tried this specific experiment, the same logic was behind our experiments using the HDAC inhibitor Trichostatin A to increase H3K27ac in SCLC. We found that HDAC inhibitors could increase this histone mark, leading to downregulation of NE factors in all SCLC cell lines, even the *EP300* mutant H2107 and H524 cells (Figure 4D). We have modified the text to better reflect this result and feel that greater exploration of this mechanism through re-expression of *EP300* is better suited for follow-up studies.

8) Related to Figure 6, did the authors test whether ETV4 is sufficient to reduce NE markers? Given that multiple ETV family members can repress NE markers, it seems plausible that it might take knockdown of all of them to rescue Kras-induced NE suppression. This could be understandably technically difficult, so perhaps the authors could just make this point clear in the text.

While we exogenously expressed ETV1 and ETV5 individually and found that they can recapitulate the effects of mutant KRAS induction on NE factor suppression in SCLC cells, we did not express ETV4. We hypothesized that, given their similar function, overexpression of anyone of these PEA3 transcription factors would phenocopy the effects of mutant KRAS induction on NE factor downregulation. Furthermore, we predicted – as highlighted by the reviewers – that it would likely require knockdown of all the ETV family members to rescue KRAS-induced NE suppression and attributed the partial rescue in suppression upon *ETV5* knockdown alone to this reasoning.

However, as ETV4 and ETV5 were the most dramatically increased upon mutant KRAS induction, we have now optimized dual knockdown to observe the combined effect on the rescue of NE factor suppression. While technically challenging as indicated by the reviewers, this experiment revealed increased rescue of NE suppression after mutant KRAS induction upon dual *ETV4*/*5* knockdown compared to knockdown of *ETV4* or *ETV5* alone. These results have been added to manuscript as Figure 6f and are further described in the text. We feel this new experimental data addresses the questions posted by the reviewers regarding the effects of ETV4.

9) Related to FiguresS8/S10, did the authors look at REST protein levels in S8 and/or S10? The functional studies for REST knockdown appear performed in only 1 of the 3 cell lines?

The reviewers bring up an important point regarding REST protein levels that we had originally tried to assess. However, despite multiple attempts using two different antibodies (Millipore [07-579] and Abcam [ab21635]), we were unable to sufficiently optimize REST detection by western blot. In the absence of protein assessment, we relied on qRT-PCR to validate the induction of *REST* after KRAS activation in SCLC as originally discovered through microarray profiling (Figure 2—figure supplement 5b). Further, we performed siRNA mediated knockdown of *REST* in H2107 cells as this cell line demonstrated the greatest increase in *REST* mRNA levels upon mutant KRAS induction as indicated by the microarray experiments. Knockdown of *REST* was confirmed by qRT-PCR and demonstrated no effect on the suppression of NE factors (Figure 2—figure supplement 5b). As no effects were observed in the line with greatest *REST* induction, we did not knockdown *REST* in additional cell lines and focused our efforts on exploring other mechanisms of action, namely the effects on chromatin modification.

10) Related to Figure S18e and S18g: This figure is confusing and could better explained in the text. In S18e, the mutation status of these genes in parental/sgControl cells prior to Osimertinib treatment should be included. Mutation status appears to be presented only after two different types of Osimertinib exposure. Second, in S18g, sgControl resistant cell lose expression of CIC protein expression entirely in the stepwise and initial exposure conditions. Consistently, sgControl Resistant cells seem to increase ETV4 and ETV5 protein levels compared to sgControl parental cells. The authors could explain whether this is due to the gain of the specific nonsense mutation. Since gain of ETV4/5 presumably inhibits NE fate conversion; does ETV knockout induce NE fate in this context?

We apologize for the confusion regarding this figure. All mutations indicated in this figure panel have been identified through comparison to the respective parental (drug naïve) cell line counterpart. Thus, only genes that have mutated in the resistant cells are indicated, as these would be predicted to potentially drive drug resistance. We have better explained this now in both the text and the respective figure legend.

For the second part of the comment, the reviewers again bring up an important potential experiment that we had previously considered. We had anticipated that knockdown of *ETV4*/*5* or re-expression of CIC in the resistant cell lines with *CIC* mutation would lead to a potential shift to a NE line state. However, our attempts to accomplish this through pharmacological means – using the pan-ETS inhibitor YK-4-279 – was complicated by a great deal of toxicity, likely due to the dependence of the resistant lines on ETV expression in the absence of ERK activity. Therefore, future experiments that circumvent this issue will be required to further explore this concept.

11) The authors may need to address certain aspects of the principal observation that the MAPK blockade resistant cells acquire NE features. Specifically, these differentiation switches typically occur in complex multiclonal populations of cancer cells in vivo, not in established cell lines. Using human PDX models or mouse models of KRAS/EGFR-driven lung cancers, the authors could have the opportunity to demonstrate the existence of such clonal shifts. Starting with in vitro established stable NE or epithelial clones could be an exceedingly difficult model in which to observe these differentiation switches. Can the authors address this point in the Discussion.

We agree that this is an important consideration that needs to be highlighted. We have now elaborated on the potential caveats of our in vitro experiments in more detail in the discussion to address this point.

[Editors' note: further revisions were suggested prior to acceptance, as described below.]

Reviewer #1:The authors largely addressed the critical points raised in the review process. While the study was designed to determine the effect of MAPK pathway activation on NE differentiation, a more clinically relevant epithelial-to-NE switch upon MAPK blockade with TKI in lung adenocarcinoma has not been modeled here, which is the limitation of this paper. With this limitation in mind, the presented work has shown several important regulatory mechanisms, namely, that hyperactive ERK signaling is a default repressor of neuroendocrine differentiation transcriptional regulators. ERK2 appears to be the dominant kinase in this process, and the downstream-induced ETV family TFs are important in executing this mechanism.1. Certain statements in the text of the paper need to be made more consistent. For example, in line 252 and below, the authors state that both, AKT and ERK are important "for the phenotypic transition", whereas the preceding discussion (line 225) states the opposite.

We apologize for this inconsistency and have reworded the second sentence to accurately reflect the results as suggested.

2. In the results and discussion, the authors should explicitly state that long-term induction with Doxy of KRAS has been toxic as early as day 7. Longer cultures (day 28) showed complete loss of KRAS suggesting a negative selection.

We have now added the following to the Results section: “This suggests negative selection of KRAS^G12V-^positive cells or epigenetic silencing of transduced KRAS^G12V^ in the long-term (28 day) culture driven by the incompatibility of KRAS activation in SCLC biology.”

3. Since many Western blot images are not statistically processed (which would be a reasonable request to the authors), explain if the figure legends whether the presented changes are representative of a certain number of repeats. It remains unclear how robust and reproducible are the results at the protein level. Certainly, concordant changes in transcriptional signatures and Westerns is a plus.

We have now added this information to the figure legends where appropriate.

4. Figure 1e presents "viability of adherent cells": please explain or correct the label to reflect exactly what was measured. If this is colony or adherent cell density based on the colony formation, then it should be stated. Unclear if the non-adherent population was simply removed in the process of staining.

We have clarified the methodology used in the representative legend for Figure 1e.

5. As relevant to most Western blot images, the Method section should state if an effort was made to retain the non-adherent population during cell collection for lysates, and if the presented lysates are representative of the total or adherent population only. This technical detail is important to interpret the results since the authors noticed growth pattern changes with introduction of MAPK-activating oncogenes.

The methods section now states that the lysates are representative of the total cell population (adherent and non-adherent).